



# Particle size-dependent fluorescence properties of water-soluble organic compounds (WSOC) and their atmospheric implications on the aging of WSOC

Juanjuan Qin [1,2], Jihua Tan [1*], Xueming Zhou [1,3], Yanrong Yang [1], Yuanyuan Qin [1], Xiaobo Wang [1], Shaoxuan Shi [1], Kang Xiao [1*], Xinming Wang [2]

[1]College of Resources and Environment, University of Chinese Academy of Sciences, Beijing, 100049, China
[2]Guangzhou Institute of Geochemistry, Chinese Academy of Sciences, Guangzhou, 510640, China
[3]Faculty of Earth Resources, China University of Geosciences, Wuhan, 430074, China

*Correspondence to*: Jihua Tan (tanjh@ucas.ac.cn); Kang Xiao (kxiao@ucas.ac.cn)

**Abstract.** Water-soluble organic compounds (WSOC) are essential in atmospheric particle formation, migration, and transformation processes. Size-segregated atmospheric particles were collected in a rural area of Beijing. Excitation-emission matrix (EEM) fluorescence spectroscopy was used to investigate the sources and optical properties of WSOC. Sophisticated data analysis on EEM data was performed to characteristically estimate the underlying connections among aerosol particles in different sizes. The WSOC concentrations and average fluorescence intensity (AFI) showed monomodal distribution in winter and bimodal distribution in summer, with dominant mode between 0.26 to 0.44 µm for both seasons. The EEM spectra of size-segregated WSOC were different among variant particle sizes, which could be the results of changing sources and/or chemical transformation of organics. Size distributions of fluorescence regional intensity (region III and V) and HIX indicate that humification degree or aromaticity of WSOC was highest between 0.26 to 0.44 µm. The Stokes shift (SS) and the harmonic mean of the excitation and emission wavelengths (WH) reflected that π-conjugated systems were high between 0.26 to 0.44 µm as well. The parallel factor analysis (PARAFAC) results showed that humic-like substances were abundant in fine particles (<1 µm) and peaked at 0.26-0.44 µm. All evidence supported that the humification degree of WSOC increased in submicron mode (<0.44 µm) and decreased gradually. Thus, it was conjectured that condensation of organics still goes on in submicron mode, resulting in the highest humification degree exhibit in particle size between 0.26 to 0.44 µm rather than <0.26µm. Synthetically analyzing 3-dimensional fluorescence data could efficiently present the secondary transformation processes of WSOC.

## 1 Introduction

The environmental, health and climate effect of aerosol particles has been reiterated for years (Pósfai and Buseck 2010; Burnett et al., 2018; Yan et al., 2020), chemical compositions and particle size are crucial for their perniciousness (Johnston and Kerecman 2019; Fan et al., 2020).



WSOC is the active fraction of organic particles, comprises 10% to 80% of organic compounds (Qin et al., 2018; Almeida et al., 2019; Cai et al., 2020). Previous researchers have proved that WSOC plays a significant role in cloud formation, solar irradiation, and atmospheric chemistry (Asa-Awuku et al., 2009; Duarte et al., 2019). However, the majority of WSOC remains mysterious, with only 10% to 20% of the organic compounds structurally identified. Generally, WSOC mixture contains both aromatic nuclei and aliphatic chains (Decesari et al., 2001; Dasari et al., 2019), incorporating with different

highly oxidized functional groups or heteroatoms like hydroxyl, carboxyl, aldehyde, ketone, amino, and other nitrogen-containing groups (Duarte et al., 2007; Cai et al., 2020). Biomass burning and secondary transformations of organics were believed to be the main sources of WSOC (ParkSeungShik et al., 2017; Xiang et al., 2017).

Many sophisticated analytical techniques have been used to unveil the chemical structural information of WSOC (Johnston and Kerecman 2019). Nuclear magnetic resonance (NMR) and mass spectrometry (MS) are two remarkable

analytical methods using to structurally unravel the complex WSOC (Duarte et al., 2020). Solution-state and solid-state $^1$H-NMR, $^{13}$C-NMR, or two-dimensional NMR are experts in obtaining structures of organics (Stark et al., 2013; Duarte et al., 2015; Chalbot et al., 2016). Mass spectrometry plays a crucial role in its high sensitivity and molecular specificity (Johnston and Kerecman 2019). Utilization of exquisite mass spectroscopy like electrospray ionization mass spectrometry (ESI-MS), ultrahigh-resolution Fourier transform ion cyclotron resonance mass spectrometry (FT-ICR-MS), and proton transfer

reaction mass spectrometry sees increasing owing to the requirement of further insight into organics in particulate matter (Cai et al., 2020; Mayorga et al., 2021). Isotopic ratio mass spectroscopy (IRMS) and accelerate mass spectroscopy (AMS) are always used to distinguish fossil combustion sources and biogenic sources by carbon isotopic (Masalaite et al., 2018; Zhao et al., 2019; Huang et al., 2020). Although having various advantages, the expanding application of formerly mentioned instruments is limited by sampling requirements or expensive costs.

Optical instruments like ultraviolet and fluorescence spectrophotometers are efficient on examination of organic functional groups in WSOC, some of them are sensitive to quantify the presence of WSOC and its sub-species like brown carbon (BrC) and humic-like substances (HULIS) (Hecobian et al., 2010; Xie et al., 2020). Light absorbing is one of the most important signatures of WSOC (Hecobian et al., 2010). Soleimanian et al., (2020) investigated the chemical properties and spectral properties of aerosol extracts found that the mass absorption efficiency (MAE) was low in the warm season and

high in the cold season. The light absorbing method could be used to observe the oxidation dynamics of WSOC for their high efficiency and sensitivity (Zhong and Jang 2014).

3-Dimensional fluorescence of excitation-emission matrix (EEM) is a sensitive and informative method that has been used in atmospheric WSOC analysis for several years (Duarte et al., 2004; Fu et al., 2014; Qin et al., 2018; Yue et al., 2019). Fluorescence analysis is a quantitative and semi-qualitative method that mainly investigating chromophoric organics like

aromatics, protein, and other organic matters containing π-conjugated systems (Xiao et al., 2018a; Xiao et al., 2020). EEM is implemented to visualize the fluorescence regions and point out possible categories of WSOC by characteristic fluorescent regions in early years (Duarte et al., 2004; Santos et al., 2009). It could reflect the aging of WSOC as well, by the red or blue shift of fluorescence peaks (Lee et al., 2013; Fu et al., 2015; Vione et al. 2019). Recent research combined fluorescence





spectrophotometry and high-resolution structural equipment to verify the chemical information digging from EEM (Chen et

al., 2016a). Fluorescence indices are important subsidiary approach to statistically analysis EEM data (Qin et al., 2018; Yue et al., 2019).

Size distributions of WSOC are explored for years (Deshmukh et al., 2016; Frka et al., 2018), the mass concentrations of WSOC generally show bimodal distributions with dominant in accumulation mode (0.05-2μm) (Yu et al., 2004; Yu et al., 2016). Structural investigations on coal burning and biomass burning affected humic-like substances (great parts of WSOC)

of four particle sizes found that organic species of all samples were basically the same without size discrepancy, but the absorption bands of aromatic groups were more intense comparing to carboxylic groups in sub-3 μm fractions (ParkSeungShik et al., 2017; Voliotis et al., 2017). Jang et al., (2019) comprehensively analyzed the structures of size-segregated humic-like substances during pre-heating, heating, and after heating periods, found that CHO, CHON, and CHOS increased in the heating period. The size distribution showed that CHO, CHON, and CHONS species decreased with particle

size and the CHOS increased, reversely, indicating the chemical structure changed with particle size.

The optical properties of size-segregated WSOC arouse increasing investigation recently. Light absorption properties of size-resolved BrC in water and methanol extracts were estimated in urban and rural Georgia, results showed that chromophores were predominant in the accumulation mode with an aerodynamic mean diameter of 0.5 μm (Liu et al., 2013). Fluorescence properties of ambient WSOC and bioaerosols in different particle sizes were estimated in a coal burning city

and a mountain site (Chen et al., 2019; Yue et al., 2019). However, enormous information is still hidden in the EEM spectra, not to mention the addition of a size-resolved stratum.

Among a group of size-segregated samples, particles of all formation status were collected in different filters, the neighbor particle sizes might share continuous transformation processes. But the relations of size-segregated WSOC were seldom analyzed before. Taking this in mind, size-segregated particles of winter and summer samples in rural Beijing were collected

to investigate the evolution of WSOC. The fast and highly efficient UV-Vis and fluorescence methods were applied in the present research, to obtain the light absorbing and fluorescent properties of size-segregated WSOC. A bunch of fluorescence indices, Stokes shift, and parallel factor analysis (PARAFAC) were performed to quantitatively disclosure the hidden connections and transformations of WSOC. Gary relational degree (GRD) is used to show the relations between particles.

## 2 Method

### 2.1 Sampling site

Size-segregated samples were collected by a 6 stage micro-orifice uniform deposit impactor (MOUDI), with aerodynamic cut-point diameters of 0.26, 0.44, 0.77, 1.4, 2.5, and 10 μm, respectively. Sample collection started at 8:00 a.m. till next 7:00 a.m. and was fixed at 23 h to reserve operating time. All samples were collected by quartz filters (Whatman) were prebaked for 5 hours (500°C) and wrapped by aluminum foil stored at -20°C after sampling.



20 groups of 6 stage size-segregated aerosol samples were collected at a rural site in Huairou Distinct, Beijing, from 14 November to 30 December 2016, and 30 June to 8 September 2017. The sample collection was non-continuous and randomly selected according to the degree of air pollution. Winter samples covered 6 levels of air quality from excellent to severe pollution day, the air quality during summer was good and moderate.

## 2.2 Chemical analysis

Organic and elemental carbon (OC and EC) were determined by thermal/optical carbon analyzer (DRI), and the thermal evolution protocol IMPROVE (Interagency Monitoring of Protected Visual Environments) was adopted. Detailed information could be found elsewhere (Cheng et al., 2009; Tan et al., 2016). The detection limit of OC and EC was 1.0 µg/m$^3$ quantified by filter and filter blank. QA and QC were performed by replicate analyses every 10 samples and the repeatability was better than 5%.

A quarter of filter sample was ultrasonically extracted twice with 5 ml ultrapure water each time and mixed up after extraction. The extract was then sifted by a 0.22 µm membrane filter to remove impurities. The measurement of WSOC was performed by a TOC analyzer (Analytic Jena AG multi N/C3100, Germany) (Xiang et al., 2017).

The extraction procedures of water-soluble ions (WSIN) were similar to WSOC but using 0.22 µm teflon filter to remove impurities. Ion chromatography (IC, Dionex ICS 900 and 1100) was used in the detection, with 8 WSIN species analyzed

($Cl^-$, $NO_3^-$, $SO_4^{2-}$, $NH_4^+$, $Na^+$, $K^+$, $Ca^{2+,}$ and $Mg^{2+}$). The recovery (90%−110%) and reproducibility (relative standard deviation of each ion lower than 5%) of the ions were implemented as well.

## 2.3 Spectrophotometer Analysis

The extraction procedures of fluorescence and ultraviolet-visible (UV-Vis) sampling were the same as WSOC detection. The excitation-emission spectra were obtained by a fluorescence spectrophotometer (F-7000, Hitachi, Japan), UV-Vis

spectra by an ultraviolet spectrophotometer (UV-2401PC, Shimadzu, Japan). To be brief, wavelength ranges are 200-400 nm for excitation and 250-500 nm for emission with 5 nm intervals (Qin et al., 2018). UV-Vis was measured with a range of 200-500 nm, 5 nm intervals. All EEM data in the present research were in Roman unit (R.U.), the background signals, interfering signals (first- and second-order Rayleigh and Raman scatterings), and the inner-filter effect were removed by subtracting an EEM of blank, replace with a band of missing values or inserting zeros outside the data area, detailed

procedures could be found in Bahram et al., (2006). Data correction and standardization followed procedures described in Xiao et al., (2016).



### 2.4 Data analysis

#### 2.4.1 Fluorescent indices

The EEM data were spectrally corrected by blank sample for instrument bias, inner filter effects, Rayleigh scattering, and
most of Raman scatter had been removed. Specific fluorescence intensity (SFI) was the fluorescence intensity divided by
WSOC concentrations. Fluorescence is an optical property determined by the chemical structure of pollutants (Andrade-
Eiroa et al., 2013a) thus fluorescent indices and partitioning methods were fairish extending to WSOC analysis.

Fluorescence indices based on intensity ratios may provide clues about the condensation state of WSOC, such as the
humification index (HIX) used to reflect the degree of humification (Kalbitz et al., 2000; Coble 2014).

$$\mathrm{HIX} = \frac{EEM_{Ex_{254},Em_{435-480}}}{EEM_{Ex_{254},Em_{300-345}}} \tag{1}$$

Fluorescence is the light emission of a substance that has absorbed light or other electromagnetic radiation. The energy
loss from fluorophore relaxing is expressed as Stokes Shift (SS), the detailed information of SS can be found in Xiao et al.,
(2019). In brief, the SS is calculated as equation (3), where $\lambda_{Ex}$ is the excitation wavelength and $\lambda_{Em}$ is the emission
wavelength. The harmonic mean of Ex/Em wavelength (WH) in equation (4) could represent the average energy level of
excited states. Thus, the SS and WH of each fluorescence intensity could be identified in an EEM spectrum.

$$SS = \frac{1}{\lambda_{Ex}} - \frac{1}{\lambda_{Em}} \tag{2}$$

$$WH = 2(\frac{1}{\lambda_{Ex}} + \frac{1}{\lambda_{Em}})^{-1} \tag{3}$$

#### 2.4.2 PARAFAC

PARAFAC model can decompose complex EEM spectra into several main components by statistical method. The
excitation spectrum, emission spectrum, and scores of each component are as follows:

$$x_{ijk} = \sum_{f=1}^{F} a_{if} b_{jf} c_{kf} + \varepsilon_{ijk}, \quad i=1,\ldots,I; \quad j=1,\ldots,J; \quad k=1,\ldots,K \tag{4}$$

Where $x$ represents the fluorescence intensity, $f$ is the number of components resolved by PARAFAC. a is proportional to
the concentration of the $f$-th component, b and c are the scaled estimating of the emission and excitation spectra. Footnote of
$i$ is the sample number, footnotes $j$ and $k$ represent emission and excitation wavelengths, respectively. Before performing
PARAFAC, all EEM data were normalized to unit norm to reducing concentration-related collinearity and avoid extremely
different leverages (Wang et al. 2020).

#### 2.4.3 Grey relational analysis (GRA)

Grey relational analysis (GRA) is a part of the grey system theory proposed by Deng (1982), that can be used to describe
the relative changes among factors in a system development process. To perform GRA analysis, references and comparison





sequences should be selected and converted to the dimensionless format. The grey relational coefficients ξ of the series and

grey relational degree are calculated as follows.

$$\xi_i(k) = \frac{\min\limits_{i}\min\limits_{k}|y(k)-x_i(k)| + \rho \max\limits_{i}\max\limits_{k}|y(k)-x_i(k)|}{|y(k)-x_i(k)| + \rho \max\limits_{i}\max\limits_{k}|y(k)-x_i(k)|} \tag{5}$$

$$GRD_i = \frac{1}{n}\sum_{k=1}^{n}\xi_i(k), \ k = 1,2,\dots,n \tag{6}$$

In which $y$ is the reference sequence and $x_i$ ($i$=1,2,3...) is the comparison sequences, $\rho$ is the distinguishing coefficient

always set as 0.5, ξ the grey relational coefficients of individual sample of the series, and $GRD_i$ is the grey relational degree

calculated by the average of $\xi_i$ (Qiu et al. 2012). The fluorescence intensity is highly affected by WSOC concentration and

many other factors, but their exact relations are not clear, thus it could be considered as a grey system and analyzed by GRA

method.

## 3 Results

### 3.1 Chemical compounds of size-segregated particles

Table **1** showed the size-segregated concentrations of WSIN, WSOC, OC, and their ratios at a rural site in Beijing for

winter and summer. WSOC showed monomodal in winter and bimodal features in summer, respectively, with a dominant

mode between 0.26 to 0.44 µm, or having a small secondary mode in particles larger than 1 µm, indicating that carbonaceous

species were mainly rich in fine particles (Huang et al., 2020). Contemporary researchers observed bimodal distribution of

organic matter in Shenzhen, China, and Gwangju, Korea (Yu et al., 2016; Huang et al., 2020).

The WSOC/OC ratios were 0.24 to 0.56 in winter and 0.16 to 0.31 in summer, which was lower than the previous records

of the polluted period in Beijing, and they were also lower than those of other cities in China (Tian et al., 2014; Wu et al.,

2020). The WSOC/OC ratios were high in fine particles with aerodynamic diameters lower than 1.4 µm and were low in

coarse mode ($PM_{2.5-10}$), which was accordant with former research on clear days in Beijing (characteristic of organic

pollution in the size-segregated aerosol Tian et al. (2016)'s results).

### 3.2 Excitation-emission spectra of size-segregated WSOC

The size-segregated EEM spectra of winter and summer WSOC were depicted in **Figure 2** (a) and (b), their fluorescence

intensities of per unit WSOC (SFI) were plotted in (c) and (d), respectively. The bulk fluorescence features of WSOC

showed evident distinctions among fine particles and coarse mode particles on EEM spectra. They differed from the spectra

of TSP samples in Japan and $PM_{2.5}$ samples in an industrial city of China, as well (Chen et al., 2016a; Qin et al., 2018),



indicating that sources affected the fluorescence properties of WSOC. The fluorescence intensities were highest in particle sizes between 0.26 to 0.44 µm during winter and summer.

Interesting characteristics were found in SFI spectra and they could reflect the fluorescence density of WSOC (Xiao et al. 2016). SFI spectra of coarse mode WSOC (which were accordant with natural sources of our unpublished research) didn't show much seasonal difference, but the SFI spectra of fine particles (matched with anthropogenic sources and secondary sources of our study) showed distinctiveness both with altering particle sizes and seasons. Especially, in winter, the SFI spectra showed a clear blue shift within region I to III when particle size increasing, while they showed humble variations in

summer.

Figure 3 was the size distributions of WSOC and average fluorescence intensity (AFI) in winter and summer, respectively. AFI showed monomodal distribution with peaks between 0.26 to 0.44 µm in winter, and bimodal distribution in summer, which was accordant with WSOC. AFI/WSOC ratios could represent the overall average fluorescence density of WSOC (Xiao et al. 2016). The AFI/WSOC ratios ranged from 0.22 to 0.57 in winter and 0.18 to 0.34 in summer, respectively. And

they were higher than that in the industrial city of Lanzhou (Qin et al. 2018). Our unpublished research found that the AFI/WSOC ratios were lower than 0.2 for anthropogenic source samples, indicating that this ratio might be higher in oxidized fluorescent WSOC.

The overall fluorescence peaks of EEM were mainly produced among regions II-V and the peaks were peak A, peak T, and peak M, which could be categorized as humic-like, tyrosine-like, and oxygenated organic substances, respectively (Qin

et al., 2018). Fluorescence regional integration (FRI) was calculated to quantify the relative strength of fluorescence intensity on regions I-V, represented by FRI1-FRI5. The FRI of each region was depicted in Figure 4. To be brief, FRI I and FRI II (protein-like) increase with particle size and peaked at coarse mode. FRI III and FRI V (HULIS) were the most abundant two fluorophores rich in fine particles. FRI IV (microbial related species) peaked between 1.4 to 2.5 µm and showed little variations with particle size increase. Almost the same size distributions of protein-like (and microbial species) and HULIS

fluorophores were also obtained by the isotopic method (Huang et al., 2020).

### 3.3 Fluorescent indices and deep properties associated with fluorescence mechanisms

Inclusive information was stored in EEM spectra, some regularities were extracted by performing division of fluorescence intensities between wavelengths. Humification index (HIX) represents the humification degree or aromaticity of fluorescent organics. Peak T/Peak C was the ratio between tryptophan and humics, suggesting the biodegradability of organics. Figure 5

showed the size distribution of HIX and Peak T/Peak C ratio in this research. HIX showed monomodal distribution and peaked between 0.26 to 0.44 µm in summer and 0.44 to 0.77 µm in winter, indicating the aromaticity of size-segregated WSOC increased firstly and decreased afterwards. Peak T/Peak C ratios of different particles increased gradually in winter, while in summer, they decreased firstly in fine particles and then increased. Peak T/Peak C peaked at coarse mode in both seasons indicating that fluorescent biogenic organics were likely to exist in larger atmospheric particles. Former researchers

also found that biogenic oxygenated organics are more inclined to sink in coarse mode (Huang et al., 2020).





Stokes shift is the energy loss of fluorophore relaxation which might associate with the π-conjected system and electron cloud density (Lakowicz, 2006). Xiao et al., (2019) found that Stokes shift of 1.2 µm$^{-1}$ is an important border of hydrophobic and hydrophilic components. High SS indicates greater energy loss due to relaxation in the excited states, organic compounds having larger π-conjugation scales are possibly exhibit high fluorescence intensity in the high SS region (Xiao et

al. 2020). The ratios of fluorescence intensity in high SS (SS>1.1) are calculated as the followed equation:

$$\eta_{SS>1.1} = \frac{\sum_{Ex}\sum_{Em} I|_{SS>1.1}}{\sum_{Ex}\sum_{Em} I} \tag{7}$$

The harmonic mean of the excitation and emission wavelengths (WH) reflects the average energy level of the excited states. On a large scale of a π-conjugated system, the electron in the ground state needs lower excitation energy jumping to the excited state (Valeur and Berberan-Santos, 2012). The ratios of fluorescence intensity in low energy state (WH>320) are

calculated as the followed equation:

$$\eta_{WH>320} = \frac{\sum_{Ex}\sum_{Em} I|_{WH>320}}{\sum_{Ex}\sum_{Em} I} \tag{8}$$

Size-segregated average SS, $\eta_{SS>1.1}$, and $\eta_{WH>320}$ were plotted in Supporting information Figure 3, and Figure 5(c). The average SS showed unobtrusive variations with particle size increase in both seasons, and $\eta_{SS>1.1}$ was slightly higher in particle sizes <1.4 µm in winter. $\eta_{WH>320}$ tended to increase between particle sizes 0.26 to 0.44 µm and decrease afterwards

both in winter and summer, indicating a larger scale of the π conjugated system or higher π-electron density in submicron particles and decreased with particle increasing.

**3.4 EEM fluorophore revealed by classification of PARAFAC results**

Parallel Factor Analysis (PARAFAC) is a mathematical method that capable of separate chemically independent but spectrally overlapping fluorescence components, on the basic assumption that EEM spectra are independent, liner related,

and additive (Murphy et al., 2011). Several prior studies have been carried out using the PARAFAC method investigating fluorescent WSOC in aerosol (Pohlker et al., 2012; Chen et al., 2019; Yue et al., 2019). Results showed that bioaerosols exhibited high bimodal signals at excitation wavelength 275nm, emission wavelength 320nm, which is sorted as protein-like organic matter. While in a typical coal burning city of China, fluorophores emerging at excitation wavelength 230-250nm and emission wavelength 380-410nm are humic-like substances with larger molecular weight.

The present study produced PARAFAC analysis separately for winter and summer samples, for the seasonal diversities of EEM spectra. Three components were extracted from winter EEM spectra: C1 was defined as HULIS-1, C2 was protein-like component and C3 was HULIS-2 (Chen et al., 2016b). However, there were just two recognizable components C1 and C2 in summer which were characterized as HULIS-1 and protein-like components, respectively. Component C3 was of no physical significance (multiple emission peak points at one single excitation wavelength) and characterized as a noise signal.

The portions of extracted components were plotted below PARAFAC results in Figure 6. Protein-like compounds were more abundant in particles larger than 2.5 µm on both seasons (37%-40% in winter and 20%-21% in summer, respectively), and HULIS showed high fractions in fine mode and was low in coarse mode particles on two seasons, which quantitatively



demonstrated that biogenic WSOC were more likely to exist in large particles and HULIS was rich in fine mode. HULIS-1 and HULIS-2 were defined in winter, their ratios HULIS-1 / HULIS-2 were low in ultrafine particles (<0.26μm) and coarse

mode, and high in fine particles with an aerodynamic diameter ranging between 0.44 to 2.5 μm, HULIS-2 was likely to be freshly emitted fluorescent WSOC and HULIS-1 exhibited fluorescent characteristics of oxidized HULIS (Vione et al. 2019).

**3.5 Specific relations among size-segregated WSOC and fluorescence properties weighted by Gary Relational Degree (GRD)**

Grey relational analysis (GRA) is suitable for solving problems with complicated interrelationships between multiple
factors and variables (Morán et al. 2006). The principle of GRA is to estimate the similarity and degree of the compactness among factors based on the geometric shape of the different sequences (Deng 1989). It has been used on multiple attribute decision-making problems (Kuo et al. 2008), evaluation of air quality control policy (You et al. 2017), and influencing factor determination of microbial product formation processes (Xu et al. 2011). GRA method was performed in the present research because atmospheric particles can be considered as a grey system for their high complexity and indeterminacy.

The GRD of size-segregated WSOC, AFI, and average UV were shown in Figure 7 (a) and (b). High GRD always refers to high connections between referencing and comparing factors. By setting particles <0.26 μm as references, (a) and (b) in Figure 7 depicted the relations among particle sizes in winter and summer, respectively. The GRD of WSOC, AFI, and UV between particle sizes were basically well among both seasons. In winter, $GRD_{0.44}$ to $GRD_{2.5}$ showed a downward tendency varying from 0.88 to 0.76 for WSOC and 0.88 to 0.78 for AFI, indicating that WSOC concentration and AFI gradually
"dislike" their original situation with particle size increasing. In summer, $GRD_{0.44}$ to $GRD_{2.5}$ showed little variations with average values at 0.64 for WSOC and 0.73 for AFI but decreased in $GRD_{10}$, indicating that WSOC in coarse mode was different from that in fine particles.

The relations between WSOC and its optical properties were showed in Figure 7 (c) and (d). AFI and average UV showed high GRD among both seasons for all particle sizes (with average GRD>0.9), indicating that fluorescence intensity and light
absorptions were closely connected with WSOC concentrations. However, clear variations of GRD were observed with particle size increasing. It was speculated that these variations were resulting from secondary transformations of WSOC because GRD were strongly negatively correlated with estimated secondary organic carbon (SOC) concentrations with correlation efficient $r$ at -0.64 ($p$<0.000) in winter and -0.63 in summer. The lowest GRD was found particle sizes between 0.26 to 0.44 μm, combining with former results of high AFI, large π-conjugation scales, and more HULIS also observed in
this particle range, it was concluded that fluorescent WSOC in particle sizes between 0.26 to 0.44 μm was highly affected by secondary processes and GRD between WSOC and AFI could serve as an indicter of secondary formation.



## 4 Discussion

By characteristically analyzing the fluorescent properties of size-segregated WSOC, the hiding relations between fluorescence and WSOC concentrations, the possible evolution of fluorescence properties during particle size increase, and

the source distinction of fluorescent WSOC in fine and coarse particles were getting clear.

Accordant with former research, the fluorescence intensities were positively related to WSOC concentrations both in winter and summer (Spearman's $r>0.8$, $p<0.001$) (Qin et al., 2018; Chen et al., 2019). The size distributions of AFI kept in step with WSOC concentrations and showed monomodal distribution in winter and bimodal distribution in summer peaked in particle sizes between 0.26 to 0.44 µm (Figure 2 (a) and (b)). The EEM spectra of size-segregated WSOC mainly exhibit

among regions 2-5 and they blue shifted with particle size increase (0.44 to 10 µm), which could be obviously observed from the EEM spectra and the increase of FRI1 and FRI2 and decrease of FRI3 and FRI5. These phenomena are explained below.

The SFI spectra (fluorescence intensity per unit WSOC) showed different properties among seasons and particle sizes. The size-segregated AFI/WSOC ratios were relatively high in fine particles with sizes between 0.26 to 1.4 µm (mainly affected by anthropogenic sources and secondary process) and low in large particles ($PM_{2.5}$), but they were both higher than

that in source samples. Freshly emitted WSOC from the source sample contains more unsaturated groups like aromatics and has lower O/C than the long transported sample (Zhang et al. 2018, Cai et al. 2020). Substitution and oxidation reactions of ambient organics might widen the delocalization of π electronics and reduce the excitation energy thereby resulting in a redshift of EEM spectra (Kalberer et al. 2004). The specific fluorescence area was widened in the ambient sample and thus having a higher AFI/WSOC ratio when WSOC concentrations at a comparable level. Continuously oxidation of organics

may break up the π system of organics and extinct fluorescence (Zanca et al. 2017). It could be inferred that ambient WSOC tending to exhibit higher AFI/WSOC ratios, while both freshly emitted WSOC and completely oxidization of WSOC could lead to a lower AFI/WSOC value.

The fluorescence indices showed vague seasonal variations. Same tendencies increased first and decreased afterwards (peaked in particle size between 0.22 to 0.44 µm) were observed in fluorescence indices of HIX and $\eta_{WH>320}$, indicating that

the π-conjugated system of WSOC increased first and decreased then. Besides, in the EEM spectra, peak M was strong in particle sizes lower than 0.77 µm and bleached in larger particles, peak A blue shifted with particle size increase. Contemporary research also found that aromatic secondary organic aerosol increased during the haze period (Yu et al. 2019).

All evidence on EEM properties and fluorescence indices above suggested that the aging of WSOC might have experienced two evolution processes of increase and decrease with particle size increasing. In the first process, fluorescence

and π-conjugated system increased and peaked between 0.22 to 0.44 µm, and peak M sparkled. Two possible pathways were proposed to explain this phenomenon. The first conjecture was heterogeneous polymerization of gas and liquid phase organics enlarged the delocalization of π electrons and leading to the increase of fluorescence (Kalberer et al. 2004). De Laurentiis et al. (2013) found that the triplet state of 1-nitronaphthalene directly reacts with phenol and forming biopolymer transformation intermediates in the liquid phase, the fluorescence spectra shifted to peak "M" during irradiation. The second





conjecture was that during oxidation processes of organics in small particles, oxygen heteroatomic rings formed or chromophoric groups like $-NO_2$ and -OH added to the fluorescent organics increased the π-conjugated system. Lee et al. (2014) observed that fluorescence increased in solar irradiation experiments of secondary organic aerosol prepared by high-$NO_x$ photooxidation of naphthalene (NAP SOA). In the second process, fluorescence decreased with particle size increasing and peak M dribbled away. This might because the further oxidation process gradually broke up the aromatic rings or

unsaturated bonds in organic matters and not fluorescent. Laboratory results also confirmed that after a long period of irradiation the fluorescence intensity of fluorescent organic decreased eventually (De Laurentiis et al. 2013, Lee et al. 2014).

PARAFAC results showed that HULIS was rich in fine particles and protein-like compounds were rich in coarse particles among both seasons, which were accordant with former research (Chen et al., 2019; Huang et al., 2020). In winter, the wavelength of HULIS-1 was slightly higher than HULIS-2 and their EEM spectra were similar to the PARAFAC results of

highly oxygenated species and less oxygenated species in Chen et al (2016b)'s study on the chromophoric WSOC. Only HULIS-1 was distinguished in summer and it could be allocated to highly oxygenated species. The sources of HULIS and protein-like compounds might be fossil fuel oxygenated organic carbon (OOC) and biogenic OOC, respectively (Huang et al., 2020).

The variations of HULIS-1 reflected that highly oxygenated species in WSOC increased first and decreased afterwards

(peaked between 0.26 to 0.44 µm), which cogently confirmed the chemical variations of WSOC happened with particle size increasing. Huang et al., (2020) observed the increase and decrease of O/C ratio in size-resolved samples in Shenzhen winter, suggesting that these variations did not rely on geographical location. The variations of HULIS-1 could also verify that secondary processes were active during the particle formation process, which confirmed the application of GRD value as an indicator of aging state of WSOC.

**5 Implications**

The SFI spectra of coarse mode WSOC were relatively stable and could serve as a referential spectrum of environmental natural sources for other research. The AFI/WSOC ratio in ambient WSOC showed vast distinction with source samples and AFI/WSOC value could be used as a potential indicator of the oxidation degree of secondary WSOC. But considering the representativeness of this ratio, more research on AFI/WSOC ratio from different sources and transformation processes

could be implemented in the future.

The variations of fluorescent characteristics in different particle sizes suggested that the fluorescence method is applicable to the research of the aging process of WSOC. Along with fluorescence indices, extensive information could be addressed in an EEM spectrum. Better understanding the connections between fluorescence property and chemical structure of organic matter, it might be possible to only use EEM data to understand the oxidation state of organics.

However, the application of EEM method still faces many uncertainties. The seasonal and particle size dependant distinctions of fluorescent WSOC suggested that the sources and transformations of anthropogenic sources were quite





different in winter and summer, secondary processes could induce fluorescence variations of WSOC. Therefore, future research could take effort to research the fluorescence characteristic of secondary WSOC.

**6 Conclusion**

In this study, a 6 stage MOUDI sampler was adopted to collected size-segregated samples of aerosol particles in a rural site of Beijing. The WSOC concentrations, UV absorption, fluorescence properties, and the energy information of fluorophores of different particle sizes were analyzed. PARAFAC method was used to decompose the mixture of fluorophores. The connections between WSOC and AFI of different particles were analyzed by grey relational degree (GRD). To sum up, the WSOC and AFI showed monomodal distributions in winter and bimodal distributions in summer. The

fluorescence efficiency (AFI/WSOC) was higher in winter comparing to summer and higher in particle sizes <1.4 µm. The variations of Fluorescence indices HIX and Peak T/Peak C ratio, and the indices reflecting energy states of fluorophores $\eta_{WH>320}$ indicated that the aromaticity or π-conjugated systems of WSOC increased in ultrafine particles (<0.44 µm) and decreased in the afterwards particle sizes. PARAFAC results showed that HULIS was rich in fine mode and protein-like sources were rich in large particles. The GRD results suggested that fluorescent WSOC in particle sizes between 0.26 to 0.44

µm were highly affected by secondary sources.

*Author contribution:* JT and KX designed the experiments, JQ collected all samples, JQ and YY carried out the experiments. JQ performed data analysis and indices calculation and KX supervised. YQ, XW and SS provided advises on data analysis and literal problems. XZ, XM, KX and JT provided technical consultations about article writing. JQ prepared the manuscript

with contributions from all co-authors.

*Acknowledge:* This work was supported by the National Natural Science Foundation of China (Grant Nos. 41675127, 41475116). We also appreciate the valuable advice from the editor who greatly improved the manuscript.

*Disclaimer Competing interests:* The authors declare that they have no conflict of interest.

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

**Table 1 Size-segregated average WSOC, WSIN concentrations, and their standard divisions.**

| | | <0.26 | 0.26-0.44 | 0.44-0.77 | 0.77-1.4 | 1.4-2.5 | 2.5-10 |
|---|---|---|---|---|---|---|---|
| Winter | $Cl^-$ | 0.42±0.25 | 1.36±1.21 | 0.83±0.72 | 1.03±0.98 | 1.19±1.27 | 0.43±0.45 |
| | $NO_3^-$ | 2.08±1.43 | 9.42±8.46 | 5.64±5.61 | 7.37±8.9 | 6.72±9.44 | 1.92±3.28 |
| | $SO_4^{2-}$ | 1.05±0.6 | 4.36±3.87 | 3.21±3.68 | 5.44±9.43 | 4.68±7.03 | 1.18±1.52 |
| | $Na^+$ | 0.12±0.05 | 0.21±0.1 | 0.16±0.08 | 0.2±0.1 | 0.52±0.6 | 0.24±0.25 |




|  |  |  |  |  |  |  |
|---|---|---|---|---|---|---|
| $NH_4^+$ | 1.05±0.57 | 2.9±2.15 | 2.05±1.82 | 2.4±2.77 | 1.67±2.18 | 0.44±0.67 |
| $Mg^{2+}$ | 0.01 | 0.01 | 0.02±0.01 | 0.05±0.04 | 0.18±0.21 | 0.08±0.09 |
| $Ca^{2+}$ | 0.06±0.01 | 0.11±0.03 | 0.15±0.08 | 0.4±0.25 | 1.67±1.35 | 0.93±0.9 |
| $K^+$ | 0.08±0.04 | 0.37±0.3 | 0.24±0.24 | 0.25±0.25 | 0.18±0.18 | 0.05±0.06 |
| OC | 4.49±1.93 | 11.04±7.2 | 5.67±4.49 | 5.45±6.26 | 5.07±3.88 | 3.4±5.17 |
| EC | 0.38±0.18 | 0.93±0.47 | 0.67±0.43 | 0.72±0.69 | 0.62±0.78 | 1.65±4.37 |
| WSOC | 1.66±0.7 | 4.73±2.96 | 2.96±2.41 | 3.21±4.33 | 2.31±2.55 | 0.64±0.5 |
| WSOC/OC | 0.38±0.07 | 0.43±0.07 | 0.56±0.27 | 0.51±0.15 | 0.37±0.14 | 0.24±0.25 |
| Summer $Cl^-$ | 0.05±0.02 | 0.1±0.04 | 0.07±0.03 | 0.07±0.02 | 0.16±0.1 | 0.11±0.06 |
| $NO_3^-$ | 0.48±0.44 | 3.5±3.32 | 1.37±1.35 | 1.04±0.86 | 4.76±4.22 | 1.49±1.37 |
| $SO_4^{2-}$ | 1.63±1.18 | 7.14±6.64 | 2.59±2.42 | 1.28±1.13 | 0.72±0.51 | 0.2±0.12 |
| $Na^+$ | 0.29±0.08 | 0.37±0.17 | 0.25±0.06 | 0.23±0.06 | 0.27±0.09 | 0.19±0.03 |
| $NH_4^+$ | 0.79±0.53 | 2.56±1.99 | 1.18±1.02 | 0.63±0.55 | 0.5±0.46 | 0.1±0.08 |
| $Mg^{2+}$ | 0.01 | 0.01 | 0.01 | 0.02±0.01 | 0.12±0.08 | 0.05±0.03 |
| $Ca^{2+}$ | 0.05±0.01 | 0.08±0.02 | 0.08±0.03 | 0.16±0.09 | 1.21±0.87 | 0.62±0.49 |
| $K^+$ | 0.03±0.02 | 0.14±0.11 | 0.05±0.04 | 0.04±0.02 | 0.06±0.02 | 0.02±0.01 |
| OC | 2.67±0.98 | 3.93±2.22 | 1.39±0.67 | 1.14±0.41 | 3.5±1.21 | 2.22±1.76 |
| EC | 0.38±0.12 | 0.44±0.16 | 0.2±0.09 | 0.22±0.06 | 0.34±0.22 | 0.5±0.52 |
| WSOC | 0.67±0.25 | 1.27±0.86 | 0.46±0.31 | 0.33±0.21 | 0.57±0.18 | 0.27±0.18 |
| WSOC/OC | 0.26±0.08 | 0.3±0.07 | 0.31±0.1 | 0.27±0.1 | 0.17±0.04 | 0.16±0.12 |

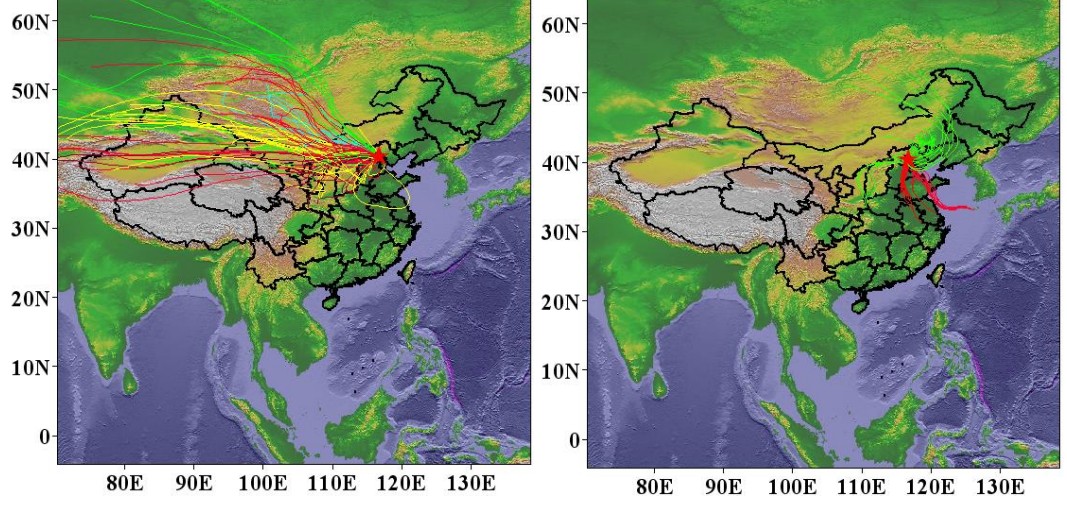



**Figure 1 The sampling site and air quality index (AQI) weighted 72 h backward trajectory of winter and summer sampling days, respectively.**

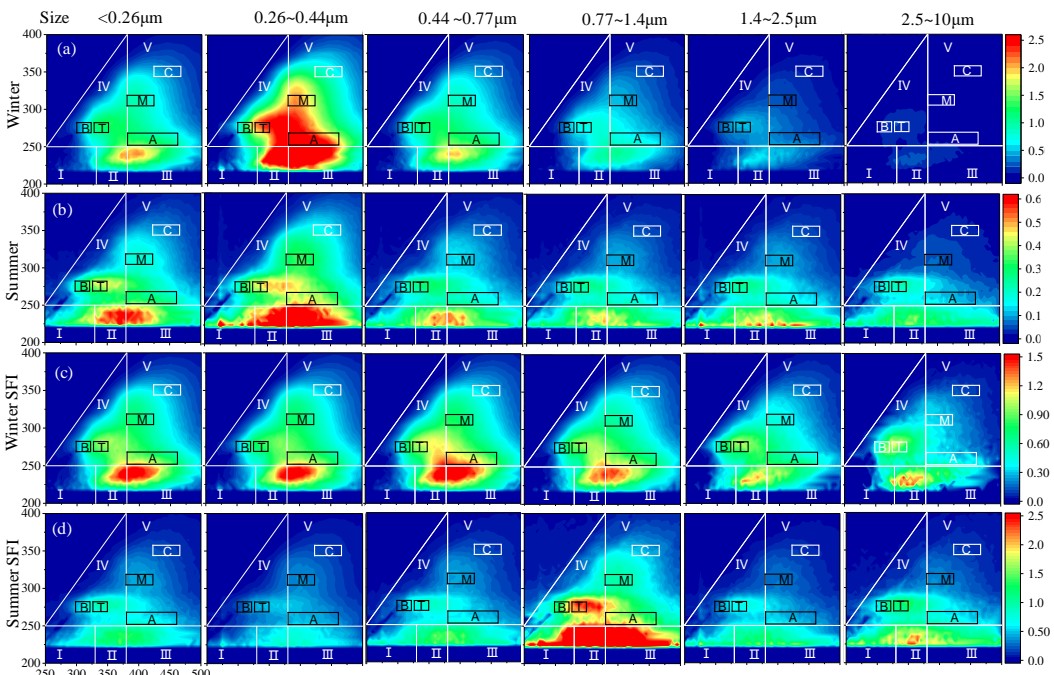

**Figure 2 Excitation-emission spectra of size-segregated samples in winter and summer. All spectra were partitioned into five regions and assigned as protein-like pollutants (I and II), fulvic acid (III), soluble microbial byproduct-like substances (IV), and humic-like acid (V), respectively (Birdwell and Engel 2010). Peak A, B, C, M, and T were generally considered as humic-like fluorophores, tyrosine-like fluorophore, humic-like carbon with larger molecular weight, marine humic-like fluorophore, and tryptophan-like fluorophores (Coble 1996). (a) and (b) were the size-segregated EEM spectra of winter and summer samples, respectively (R.U.), (c), and (d) were the corresponding EEM spectra of fluorescence emitted per unit of WSOC.**



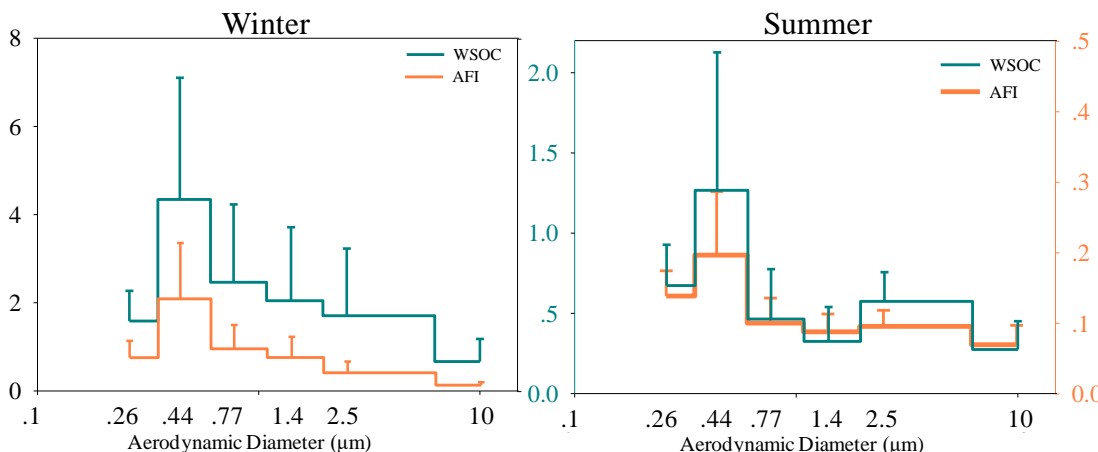

**Figure 3 Size distributions of WSOC and AFI in winter and summer. AFI was in Roman unit.**

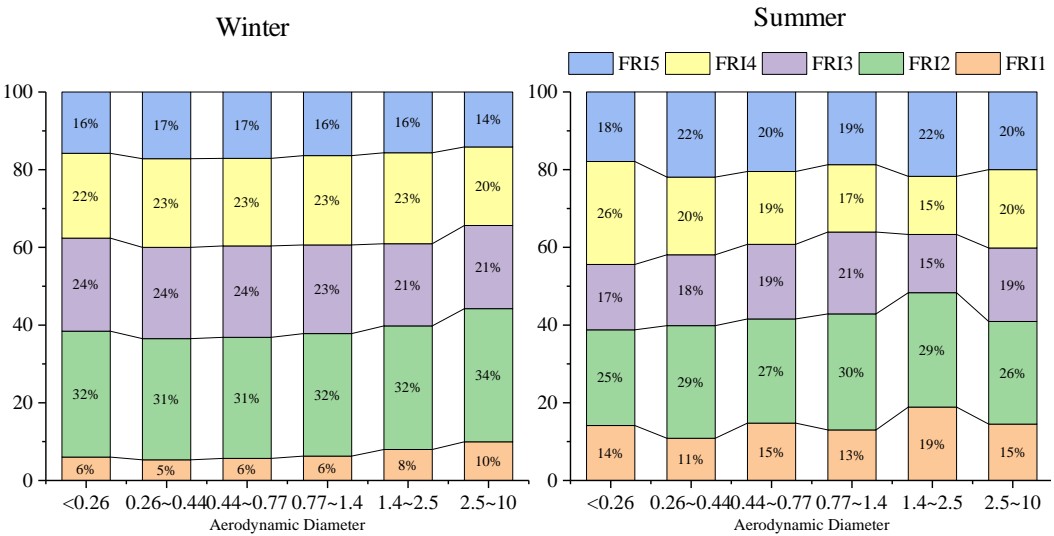

**Figure 4 Size distribution of fluorescence regional intensity for winter and summer. FRI1-FRI5 was FRI of fluorescence region I to V.**

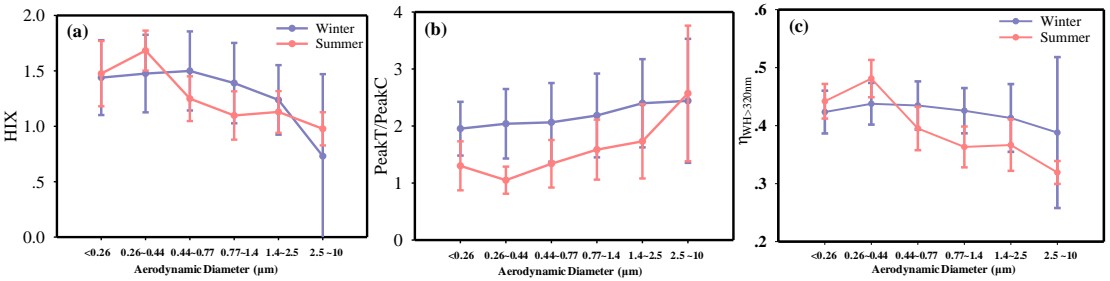



**Figure 5 Humic index and Peak T/Peak C ratio served as indictors of humification degree and the biodegradable possibility of WSOC. (a) HIX in different particle sizes, large HIX value indicated high humification degree or high aromaticity of fluorescent organics. (b) Peak T/Peak C ratios of different particle sizes. The large value indicated more microbial metabolites in the fluorescent organics. (c) showed the size distributions of η$_{WH>320}$ for winter and summer samples, respectively.**

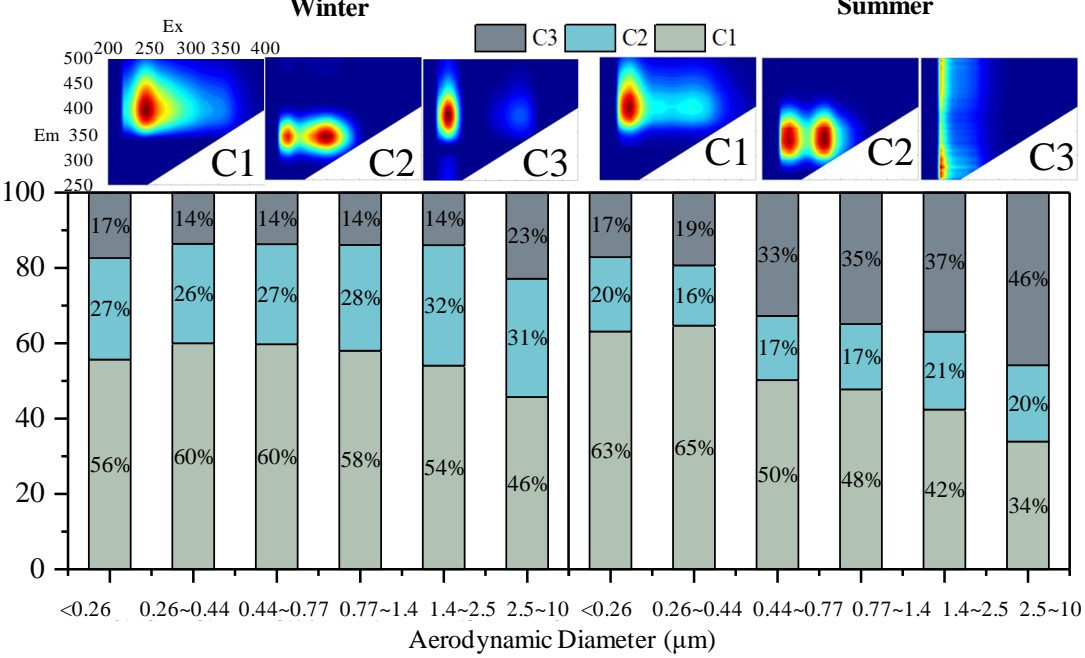


**Figure 6 PARAFAC results of EEM in winter and summer respectively. Three components were extracted of both seasons, the portions of each component for different particle sizes were shown as well.**





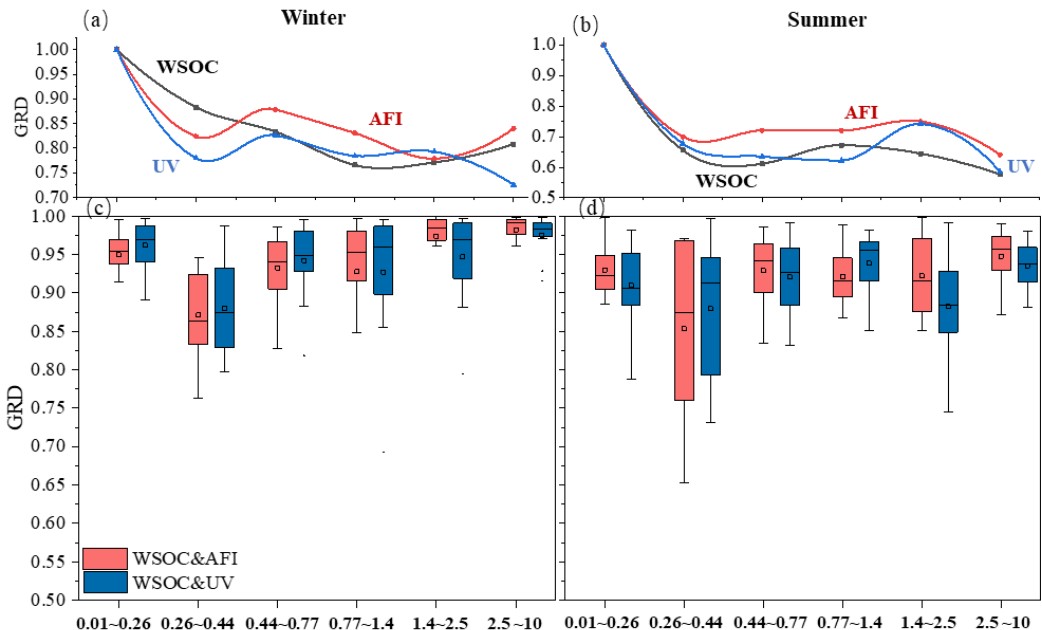

**Figure 7 Grey relational degree (GRD) of size-segregated WSOC, AFI, and average UV. (a) and (b) GRD calculated**

**by WSOC, AFI, and average UV of each sample, setting data of 0.01 to 0.26 as references, GRD (0.01-0.26) =1; (c)**

**and (d) GRD between WSOC and light absorption indices, setting WSOC as references.**