# Peer review of "Particle size-dependent fluorescence properties of water-soluble organic compounds (WSOC) and their atmospheric implications on the aging of WSOC"

_Atmospheric Chemistry and Physics, 2021_

## Referee Comment (RC5)

**General Comments**: This manuscript titled "Particle size-dependent fluorescence properties of water-soluble organic compounds (WSOC) and their atmospheric implications on the aging of WSOC" describes fluorescence properties of size-segregated WSOC aerosols in a rural area of Beijing. To attain the study objectives, the authors applied different data analysis tools to the excitation-emission matrix (EEM) fluorescence spectra of the size segregated WSOC. The WSOC aerosols represent a significant fraction of organic aerosols and one of the driving factors in climate change due to their light absorption properties. The topic of the study is within the scope of the journal and has relevance to the atmospheric research community. Although the technique used (i.e., EEM fluorescence spectra) in this study might have some limitations (such as difficulties in segregating anthropogenic and bioaerosols WSOC fractions etc) if used alone. However, size-segregated EEM fluorescence spectra (this study) can be helpful in mitigating many of these limitations and understanding anthropogenic and natural sources of WSOC, their atmospheric evolution, and optical properties. Even so, this study has many shortcomings in its current version given below:

**Major Comments:**

Introduction: Lacks organization and continuity? The reviewer suggests the revision of introduction section to make it more organized and in tandem with the objective of the study.

Line 161: The size distribution of WSOC/OC and WSOC concentration doesn't follow similar trend. Although several studies in the past (Dasari et al., 2019 science advances; Choudhary et al., 2021 environmental pollution) as well as this study (in introduction) have stated that majority of WSOC are secondary (oxidized) in nature. The author can elaborate possible rationales briefly?

Line 174: Figure 1 is not discussed in the manuscript. Either delete it or add some relevant discussion about same?

Line 186-193 and 285-290: The author stated that "The AFI/WSOC ratios ranged from 0.22 to 0.57 in winter and 0.18 to 0.34 in summer, respectively." "Our unpublished research found that the AFI/WSOC ratios were lower than 0.2 for anthropogenic source samples, indicating that this ratio might be higher in oxidized fluorescent WSOC." If that is the case, size distribution of AFI/WSOC should have follow the distribution trend of WSOC/OC (a tracer for photochemical oxidation), but this is not the case in this paper (Figure 3). Explain the rationale/s behind this behaviour?

Line 212: The author stated that Stokes shift (SS) of 1.2 µm-1 is an important border of hydrophobic and hydrophilic components. And later used Stokes shift of 1.1 to determine ratios of fluorescence intensity in high SS. Elaborate the possible reason/s?

Line 205: HIX (aromaticity) and WSOC/OC (oxidation) ratio following same size distribution trend. How come? This could be an important finding of the manuscript. Add some discussion about same in Discussion and Implication sections.

Line 209, 210, 243: The author categorized Protein-like compounds into biogenic origin. But aerosols partitioned from VOCs (isoprene etc.) emitted from plants also categorized into biogenic aerosols. Does the author also incorporating these aerosols produced from VOCs in Protein-like compounds or it is just bioaerosols? Please clarify?

Line 256: Why did author used particles <0.26 µm as references for Grey relational analysis (GRA)? Why not use size bins where WSOC, UV and AFI are maximum?

**Minor Corrections:**

In the Reviewer's opinion, the English language needs significant revision throughout the manuscript before acceptance. The some of English-related corrections and other minor

comments are suggested below:

Line 33: Replace "mysterious" with either "Unknown" or "Uncharacterized".

Line 34-35: The sentence lacks continuity. Revise the sentence "incorporating with different highly oxidized functional groups or heteroatoms like" with may be something like "WSOC mixture contains both aromatic nuclei and aliphatic chains (Decesari et al., 2001; Dasari et al., 2019), with functional groups or heteroatoms like hydroxyl, carboxyl, aldehyde, ketone, amino, and other nitrogencontaining groups (Duarte et al., 2007; Cai et al., 2020)".

Line 37: Is the reference "(ParkSeungShik et al., 2017)" is correctly cited and listed in the reference list (also see line 485).

Line 39: Revise "Nuclear magnetic resonance (NMR) and mass spectrometry (MS) are two remarkable analytical methods using to structurally unravel the complex WSOC (Duarte et al., 2020)."

line 46: It is "Accelerator" not "accelerate".

Line 46: Revise the sentence to something like "Isotopic ratio mass spectroscopy (IRMS) and accelerator mass spectroscopy (AMS) are widely used to distinguish organic emissions from fossil combustion sources and biogenic sources using carbon isotopic characteristics (Masalaite et al., 2018; Zhao et al., 2019; Huang et al., 2020)."

Line 50-56: Whole paragraph lacks organization and continuity. The reviewer suggests the revision of the paragraph.

Line 57: Replace "3-Dimensional fluorescence of excitation-emission matrix (EEM)" to "3-Dimensional excitation–emission matrix (EEM) fluorescence spectroscopy"

Line 59: it should be "mainly helpful in investigating"

Line 62: what does author mean by "in early years"? Does author mean "earlier studies", if so, revise the sentense.

Line 65: It should be "analyse" not "analysis"

Line 69: "(great parts of WSOC)"? It should be something like "significant fraction of WSOC"

Line 70: "reversely"?

Line 82: "neighbor particle sizes" should "adjacent particle size bins"

Line 83: The use of "But" is not perfect here. The reviewer suggests to use "and" instead.

Line 94: confusing sentence "All samples were collected by quartz filters (Whatman) were prebaked for 5 hours (500°C) and wrapped by aluminum foil stored at -20°C after sampling." May be revised to "All samples collected on quartz filters (Whatman), prebaked for 5 hours (500°C) before sample collection, were wrapped by aluminum foil after sampling and stored at -20°C."

Line 95: Need clarification? Total 20 groups for 2 seasons or 20 groups each for every season?

Line 106: Should be "The extract was then filtered through a 0.22 μm membrane filter to remove impurities."

Line 113: Confusing? The sentence may be written like "The extraction procedure of samples subjected to fluorescence and ultraviolet-visible (UV-Vis) measurements were same as WSOC detection."

Line 117: Should be "Raman Unit"

Line 124: Revise the sentence "The EEM data were spectrally corrected by blank sample for instrument bias, inner filter effects, Rayleigh scattering, and

most of Raman scatter had been removed" to "The EEM data were spectrally corrected by blank sample to remove interferences from instrument bias,  inner filter effects, Rayleigh scattering, and Raman scatter."

Line 133-134: Equations number is not matching? Example: "equation (3)" should be "equation (2)" and "equation (4)" should be "equation (3)"

Line 218: Revise "On a large scale of a π-conjugated system, the...."

Line 222: "Supporting information Figure 3, and Figure 5(c)." shoud be Figure S3 and Figure S5(c). Do same thing for Figures S1, S2, S4 and Table S1, in Supporting Information.

Line 87 and 228: The full form of PARAFAC is already mentioned on Line 87. No need to repeat it again. Follow same comment for others as well (e.g. GRA on line 249 etc.).

---

## Author Comment (AC3)

Dear reviewer:

Thank you very much for your valuable advice on our present research name as "Particle size-dependent fluorescence properties of water-soluble organic compounds (WSOC) and their atmospheric implications on the aging of WSOC", we have addressed all comments carefully, and the detailed corrections are described in the later paragraph. We also have sent our revised manuscript to professionals and native speakers to polish the language, we believe the language of the next version will be vastly improved. Thank you again for your time and patience.

Major Comments:

1. **Introduction:** Lacks organization and continuity? The reviewer suggests the revision of the introduction section to make it more organized and in tandem with the objective of the study.

**Thank you very much for your valuable advice.** We have reorganized the introduction section into four parts: Paragraph one is the general topic of WSOC; paragraphs two and three introduce several advanced analytical methods that have been used in recent research and refers their limitations, then the advantage of optical methods is proposed; paragraph four lists several recent research on size segregated WSOC; and the perspective of the present research is summarized in the end. The new introduction is shown as follows.

"The environmental, health and climate effect of aerosol particles has been reiterated for years (Pósfai and Buseck 2010; Burnett et al., 2018; Yan et al., 2020; Fan et al., 2020). WSOC is the active fraction of aerosol particles, comprises 10% to 80% of organic compounds (Qin et al., 2018; Almeida et al., 2020; Cai et al., 2020). Previous researchers have proved that WSOC plays a significant role in cloud formation, solar irradiation, and atmospheric chemistry (Asa-Awuku et al., 2009; Duarte et al., 2019). However, the majority of WSOC remains uncharacterized, with only 10% to 20% of the organic compounds structurally identified. Generally, WSOC mixture contains both aromatic nuclei and aliphatic chains (Decesari et al., 2001; Dasari et al., 2019), with functional groups or heteroatoms like hydroxyl, carboxyl, aldehyde, ketone, amino, and other nitrogen-containing groups (Duarte et al., 2007; Cai et al., 2020). Biomass burning and secondary transformations of organics were believed to be the main sources of WSOC (Park et al., 2017; Xiang et al., 2017).

Many sophisticated analytical techniques have been used to unveil the chemical structural information of WSOC (Johnston and Kerecman 2019). Nuclear magnetic resonance (NMR) are experts in obtaining structures of organics (Stark et al., 2013; Duarte et al., 2015, 2020; Chalbot et al., 2016). Mass spectrometry plays a crucial role in its high sensitivity and molecular specificity. The electrospray ionization with ultrahigh-resolution Fourier-transform ion cyclotron resonance mass spectrometry (ESI-FT-ICR-MS), and proton transfer reaction mass spectrometry (PTR-MS) sees increasing application because of the requirement of further insight into organics in particulate matter (Cai et al., 2020; Mayorga et al., 2021). Isotopic ratio mass spectroscopy (IRMS) and accelerator mass spectroscopy (AMS) are widely used to distinguish organic emissions from fossil combustion sources and biogenic sources using carbon isotopic characteristics (Masalaite et al., 2018; Zhao et al., 2019; Huang et al., 2020).

Although having various advantages, the expanding application of formerly mentioned instruments is limited by sampling requirements or expensive costs. In contrast, optical instruments like ultraviolet and fluorescence spectrophotometers are relatively low-cost and efficient. Moreover, the results of the optical method can provide quantitative and qualitative information simultaneously,

which guaranteed their broad application on the research of organics like dissolved organic matter (DOM) in water and WSOC (Hecobian et al., 2010; Qin et al., 2018; Xiao et al., 2016). 3-Dimensional excitation-emission matrix (EEM) fluorescence spectroscopy is an informative optical method that has been used in atmospheric WSOC analysis (Duarte et al., 2004; Fu et al., 2014). Fluorescence analysis is helpful in investigating chromophoric organics like aromatics, protein, and other organic matters containing π-conjugated systems (Xiao et al., 2018; 2020). EEM spectrum is implemented to visualize the fluorescence regions and point out possible categories of WSOC by characteristic fluorescent regions in earlier studies (Duarte et al., 2004; Santos et al., 2009). It could reflect the aging of WSOC as well, by the red or blue shift of fluorescence peaks (Lee et al., 2013; Fu et al., 2015; Vione et al., 2019). Fluorescence indices are important subsidiary approaches to statistically analysis EEM data (Qin et al., 2018; Yue et al., 2019), which were determined by the chemical structure of pollutants (Andrade-Eiroa et al., 2013a).

Size distributions of WSOC have been explored for years (Deshmukh et al., 2016; Frka et al., 2018), and the optical properties of size segregated WSOC arouse increasing investigation recently. Generally, the mass concentrations of WSOC show bimodal distributions with dominant in accumulation mode (0.05-2μm) (Yu et al., 2004; Yu et al., 2016). Structural investigations on coal burning and biomass burning affected humic-like substances (a significant fraction of WSOC) of four particle sizes found that organic species of all samples were the same without size discrepancy, reversely, the absorption bands of aromatic groups were more intense compared to carboxylic groups in sub-3 μm fractions (Park et al., 2017; Voliotis et al., 2017). Jang et al., (2019) comprehensively analyzed the structures of size segregated humic-like substances during pre-heating, heating, and after heating periods, found that the chemical structure of HULIS changed with particle size. Liu et al., (2013) examined the light absorption properties of size-resolved BrC and methanol extracts in Georgia, results showed that chromophores were predominant in the accumulation mode with an aerodynamic mean diameter of 0.5 μm. More recently, even fluorescence properties for size segregated ambient WSOC and bioaerosols have been estimated in a coal burning city and a mountain site (Chen et al., 2019; Yue et al., 2019).

Yet comprehensively analyzing fluorescence properties for size-resolved aerosol is infrequent, enormous information is still hidden in the EEM spectra, not to mention the adjacent particle size bins. The perspective of the present research is to investigate the fluorescence properties of WSOC in different particle sizes and try to shed light on the size-dependent evolution of WSOC. 6 stage size segregated particles samples of winter and summer were collected in rural Beijing. The fast and efficient UV-Vis and fluorescence methods were applied in the present research, to obtain the light-absorbing and fluorescent properties of size segregated WSOC. A bunch of fluorescence indices, Stokes shift, and PARAFAC were performed to quantitatively disclosure the hidden connections and transformations of WSOC. Gary relational degree (GRD) is used to show the relations between particles."

2. Line 161: The size distribution of WSOC/OC and WSOC concentration doesn't follow similar trend. Although several studies in the past (Dasari et al., 2019 science advances; Choudhary et al., 2021 environmental pollution), as well as this study (in introduction), have stated that majority of WSOC are secondary (oxidized) in nature. The author can elaborate possible rationales briefly?

**Thank you very much for the question.** We are sorry for not discussing the size distribution of WSOC/OC ratios and WSOC. As a result, the size distribution of WSOC/OC and WSOC

concentration doesn't follow a similar trend indeed. We can see in Figure A. below, that WSOC and OC show similar tendencies for both seasons, and the peaks of WSOC/OC show a delay (peaks around 1um) comparing to their concentrations. This may be because of an increased portion of WSOC or a decrease of OC, as we all know, WSOC is part of OC, thus, we prefer to believe that more OC oxidized to WSOC in particle sizes around 1um, since particles of those sizes can long-exist in the atmospheric environment.

Another possible reason might be that the difference in sources and transformation process of size segregated particles might lead to multiple WSOC/OC results. Ram et al. (2012) reported the WSOC/OC ratio of 0.47±0.11, and characteristically listed the former reported results in vehicle exhaust aerosol and biomass burning affected aerosol, and found that the ambient aerosol had a higher WSOC/OC value and concluded that WSOC/OC could serve as an indicator of secondary formation.

Reference:
Ram, K., Sarin, M. M., and Tripathi, S. N.: Temporal trends in atmospheric PM(2).(5), PM(1)(0), elemental carbon, organic carbon, water-soluble organic carbon, and optical properties: impact of biomass burning emissions in the Indo-Gangetic Plain, Environ Sci Technol, 46, 686-695, 10.1021/es202857w, 2012.

[Figure]

Figure A. size distributions of WSOC, OC and WSOC/OC

3. Line 174: Figure 1 is not discussed in the manuscript. Either delete it or add some relevant discussion about same?

**Thank you very much for your valuable advice.** We are sorry for not clarify the intention of putting Figure 1 in the manuscript. It is the AQI (air quality index) weighted 72h backward trajectories of our sampling period, providing the sampling information of present research. We have referred it in the manuscript section 2.1.

Lines 98-99: "The air quality index weighted 72h backward trajectories during the sampling period were exhibited in 错误!未找到引用源。." 69671251

4. Line 186-193 and 285-290: The author stated that "The AFI/WSOC ratios ranged from 0.22 to 0.57 in winter and 0.18 to 0.34 in summer, respectively." "Our unpublished research found that the AFI/WSOC ratios were lower than 0.2 for anthropogenic source samples, indicating that this ratio might be higher in oxidized fluorescent WSOC." If that is the case, size distribution of AFI/WSOC should have follow the distribution trend of WSOC/OC (a tracer for photochemical oxidation), but this is not the case in this paper (Figure 3). Explain the rationale/s behind this behaviour?

**Thank you very much for your valuable advice.** We also have noticed this inappropriate

deduction during the revision period, thus the indication sentences of "..., indicating that this ratio might be higher in oxidized fluorescent WSOC." have been deleted in line 193. Because the phenomenon of low AFI/WSOC values in sources samples and $PM_{2.5}$ samples in a polluted city and relatively high values in ambient environment samples cannot strongly confirm this conjecture, and we provide some possible explanations instead in the Discussion section. The different size distributions of AFI/WSOC and WSOC/OC may be because that oxidization of WSOC causes fluorescence quenching, we mentioned this perspective in lines 295-298. We have reconstructed the sentences in lines 186-193 and 285-290 as follows.

Lines 184-187 "AFI/WSOC ratios could represent the overall average fluorescence density of WSOC (Xiao et al., 2016). Our unpublished research found that the AFI/WSOC ratios were lower than 0.2 for anthropogenic source samples. And in the present research, the AFI/WSOC ratios ranged from 0.22 to 0.57 in winter and 0.18 to 0.34 in summer, respectively, which were higher than that in source samples and the industrial city of Lanzhou (Qin et al., 2018)."
Lines 292-298 "Substitution and oxidation reactions of ambient organics might widen the delocalization of $\pi$ electronics and reduce the excitation energy thereby resulting in a redshift of EEM spectra (Kalberer et al., 2004). The specific fluorescence area was widened in the ambient sample and thus having a higher AFI/WSOC ratio when WSOC concentrations were at a comparable level. However, continuously oxidation of organics may break up the $\pi$ system of organics and extinct fluorescence (Zanca et al., 2017), and lead to a relatively low AFI/WSOC value for the particles having a long residence time, thus the AFI/WSOC values showed a wide variance for different particles."

5. Line 212: The author stated that Stokes shift (SS) of 1.2 μm-1 is an important border of hydrophobic and hydrophilic components. And later used Stokes shift of 1.1 to determine ratios of fluorescence intensity in high SS. Elaborate the possible reason/s?
**Thank you very much for the question.** We are sorry for not explaining the reason for using SS of $1.1 um^{-1}$ as the border of hydrophobic and hydrophilic components. As a fact, Xiao's former research found that hydrophobic fractions tended to present fluorescence peaks at $SS>1 um^{-1}$, thus we use the average value of SS at $1.1 um^{-1}$ as the border, we add the description in line 224. The context is now shown as follows.
"Xiao et al., (2019) found that of stokes shift near 1.2 μm$^{-1}$ is an important border of hydrophobic and hydrophilic components. Hydrophobic fractions tend to have higher intensity in stokes shifts >1.2, possibly as a result of the larger scale of the $\pi$ conjugated system, versus, hydrophilic contents usually have ionogenic groups bond with fluorescent aromatics reduced the π-conjugated systems, hence, leading to high fluorescence intensities sitting on both sides of stokes shifts around 1.2. While their earlier research also reported that hydrophobic fractions tended to present fluorescence peaks at Stokes shift >1 (Xiao et al., 2016). Thus, the ratios of fluorescence intensity in high SS (SS>1.1) are calculated as the followed equation:"

6. Line 205: HIX (aromaticity) and WSOC/OC (oxidation) ratio following same size distribution trend. How come? This could be an important finding of the manuscript. Add some discussion about same in Discussion and Implication sections.
**Thank you very much for your valuable advice.** We have compared the size distributions of HIX

and WSOC/OC ratio, find that they both show monomodal distributions and the peaks of WSOC/OC delay to larger particle sizes comparing to HIX. That means the HIX starts to decrease when WSOC/OC still increases. The decrease of HIX maybe because the fluorescent WSOC is oxidized to non-fluorescent organics during a long period of exposure in the ambient environment, as the fluorescence property requires conjugated systems in organics. While the long oxidation period also leads to more OC convert to WSOC at the same time, thus the WSOC/OC ratio keeps increase. We have added some discussion in lines 304-307 as follows.

Lines 304-307 "Besides, it was noticed that HIX and WSOC/OC showed similar size distributions except for larger peaking particle size of WSOC/OC values comparing to HIX. This might be because the fine particles with relatively large sizes could long exist in atmospheric oxidation environment, the WSOC/OC ratios increase gradually, however, the oxidation process could also cause fluorescence quenching and lead to the decrease of HIX (Vione et al., 2019)."

7. Line 209, 210, 243: The author categorized protein-like compounds into biogenic origin. But aerosols partitioned from VOCs (isoprene etc.) emitted from plants also categorized into biogenic aerosols. Does the author also incorporating these aerosols produced from VOCs in Protein-like compounds or it is just bioaerosols? Please clarify?

**Thank you very much for your valuable advice.** We are sorry for the unclear description of biogenic sources. In the present research, we have only considered protein-like compounds in the particulate organics as biogenic and neglected that biogenic VOC are biogenic aerosols as well. To avoid misunderstanding, we have changed the "biogenic sources" to "microbial related" throughout the article. The "biogenic oxygenated organics" in line 215 is a term of Huang's research, thus we keep it unchanged. The other corrections are as follows.

Line 214: "Peak T/Peak C peaked at coarse mode in both seasons indicating that fluorescent microbial related species were likely to exist in larger atmospheric particles."

Line 256: "demonstrated that microbial related WSOC were more likely to exist in large particles"

8. Line 256: Why did author used particles <0.26 μm as references for Grey relational analysis (GRA)? Why not use size bins where WSOC, UV and AFI are maximum?

**Thank you very much for the question.** We are sorry for not carefully explain the reasons for reference list selection. The minimum particle size is selected because it is assumed that the increase of particle size is an accumulation process, we are trying to find the connections between different particle sizes during the particle increase processes. We have added some explanations in Section 2.4.3, they are shown as follows.

Lines 154-157: "Firstly, considering the evolution of particle sizes as a changing system, larger particles might come from the accumulation and transformation of smaller particles, especially for ultrafine particles. By setting data of particles smaller than 0.26 μm (WSOC concentrations, AFI or UV) as references and particles larger than 0.26 μm as comparisons, their affinities were analyzed by GRA."

Minor Corrections:

In the Reviewer's opinion, the English language needs significant revision throughout the manuscript before acceptance. The some of English-related corrections and other minor comments

are suggested below:

**We are sorry for the mistakes in the last version of the manuscript.** We have carefully addressed all of the suggestions and corrected them, we also have checked through the article and modified them.

1. Line 33: Replace "mysterious" with either "Unknown" or "Uncharacterized".

**Thank you very much for your valuable advice.** We have changed "mysterious" to "Uncharacterized" in lines 31-32, they are now shown as follows.

"However, the majority of WSOC remains uncharacterized, with only 10% to 20% of the organic compounds structurally identified."

2. Line 34-35: The sentence lacks continuity. Revise the sentence "incorporating with different highly oxidized functional groups or heteroatoms like" with may be something like "WSOC mixture contains both aromatic nuclei and aliphatic chains (Decesari et al., 2001; Dasari et al., 2019), with functional groups or heteroatoms like hydroxyl, carboxyl, aldehyde, ketone, amino, and other nitrogencontaining groups (Duarte et al., 2007; Cai et al., 2020)".

**Thank you very much for your valuable advice.** We have revised this sentence as follows.

Line 32-34 "Generally, WSOC mixture contains both aromatic nuclei and aliphatic chains (Decesari et al., 2001; Dasari et al., 2019), with functional groups or heteroatoms like hydroxyl, carboxyl, aldehyde, ketone, amino, and other nitrogen-containing groups (Duarte et al., 2007; Cai et al., 2020)."

3. Line 37: Is the reference "(ParkSeungShik et al., 2017)" is correctly cited and listed in the reference list (also see line 485).

**Thank you very much for your valuable advice.** We have checked the reference list and modified the citation format in the text. The citations are shown as follows.

In the text "(Park et al., 2017)"

In the reference list "Park S., Yu, J., Yu, G.-H. and Bae M. S.: Chemical and absorption characteristics of water-soluble organic carbon and humic-like substances in size segregated particles from biomass burning emissions, Asian J. Atmos. Environ., 11, 96-106, https://doi.org/10.5572/ajae.2017.11.2.096, 2017."

4. Line 39: Revise "Nuclear magnetic resonance (NMR) and mass spectrometry (MS) are two remarkable analytical methods using to structurally unravel the complex WSOC (Duarte et al., 2020)."

**Thank you very much for your valuable advice.** We have reconsidered the context and modified the whole paragraph. This sentence is deleted now.

5. line 46: It is "Accelerator" not "accelerate".

**Thank you very much for your valuable advice.** Because we have reconstructed and modified the introduction section, this sentence is deleted now.

6. Line 46: Revise the sentence to something like "Isotopic ratio mass spectroscopy (IRMS) and accelerator mass spectroscopy (AMS) are widely used to distinguish organic emissions from fossil combustion sources and biogenic sources using carbon isotopic characteristics (Masalaite et al.,

2018; Zhao et al., 2019; Huang et al., 2020)."

**Thank you very much for your valuable advice.** We have corrected the sentence as suggested. They are now shown as follows in lines 42-44.

"Isotopic ratio mass spectroscopy (IRMS) and accelerator mass spectroscopy (AMS) are widely used to distinguish organic emissions from fossil combustion sources and biogenic sources using carbon isotopic characteristics (Masalaite et al., 2018; Zhao et al., 2019; Huang et al., 2020)."

7.    Line 50-56: Whole paragraph lacks organization and continuity. The reviewer suggests the revision of the paragraph.

**Thank you very much for your valuable advice.** We have reconstructed the paragraph as follows.

"Although having various advantages, the expanding application of formerly mentioned instruments is limited by sampling requirements or expensive costs. In contrast, optical instruments like ultraviolet and fluorescence spectrophotometers are relatively low-cost and efficient. Moreover, the results of the optical method can provide quantitative and qualitative information simultaneously, which guaranteed their broad application on the research of organics like dissolved organic matter (DOM) in water and WSOC (Hecobian et al., 2010; Qin et al., 2018; Xiao et al., 2016). 3-Dimensional excitation-emission matrix (EEM) fluorescence spectroscopy is an informative optical method that has been used in atmospheric WSOC analysis (Duarte et al., 2004; Fu et al., 2014). Fluorescence analysis is helpful in investigating chromophoric organics like aromatics, protein, and other organic matters containing $\pi$-conjugated systems (Xiao et al., 2018; 2020). EEM spectrum is implemented to visualize the fluorescence regions and point out possible categories of WSOC by characteristic fluorescent regions in earlier studies (Duarte et al., 2004; Santos et al., 2009). It could reflect the aging of WSOC as well, by the red or blue shift of fluorescence peaks (Lee et al., 2013; Fu et al., 2015; Vione et al., 2019). Fluorescence indices are important subsidiary approaches to statistically analysis EEM data (Qin et al., 2018; Yue et al., 2019), which were determined by the chemical structure of pollutants (Andrade-Eiroa et al., 2013a)."

8.    Line 57: Replace "3-Dimensional fluorescence of excitation-emission matrix (EEM)" to "3-Dimensional excitation–emission matrix (EEM) fluorescence spectroscopy"

**Thank you very much for your valuable advice.** We have corrected the phrase as suggested in lines 49-50.

"3-Dimensional excitation-emission matrix (EEM) fluorescence spectroscopy is an informative optical method that has been used in atmospheric WSOC analysis."

9.    Line 59: it should be "mainly helpful in investigating"

**Thank you very much for your valuable advice.** We have corrected the sentence in line 51.

"Fluorescence analysis is helpful in investigating chromophoric organics"

10.    Line 62: what does author mean by "in early years"? Does author mean "earlier studies", if so, revise the sentense.

**Thank you very much for your valuable advice.** we have modified the phrase in lines 52-54 as follows.

"EEM spectrum is implemented to visualize the fluorescence regions and point out possible

categories of WSOC by characteristic fluorescent regions in earlier studies (Duarte et al., 2004; Santos et al., 2009)"

11.   Line 65: It should be "analyse" not "analysis"
**Thank you very much for your valuable advice.** Because we have reconstructed and modified the introduction section, this sentence is deleted now.

12.   Line 69: "(great parts of WSOC)"? It should be something like "significant fraction of WSOC"
**Thank you very much for your valuable advice.** We have corrected the phrase accordingly as follows.
"Structural investigations on coal burning and biomass burning affected humic-like substances (a significant fraction of WSOC) of four particle sizes"

13.   Line 70: "reversely"?
**Thank you very much for your valuable advice.** We have changed "but" to "reversely" in line 62, they are now shown as follows.
"reversely, the absorption bands of aromatic groups were more intense compared to carboxylic groups in sub-3 μm fractions (Park et al., 2017; Voliotis et al., 2017)."

14.   Line 82: "neighbor particle sizes" should "adjacent particle size bins"
**Thank you very much for your valuable advice.** We have modified the phrase in line 70.
"Yet comprehensively analyzing fluorescence properties for size-resolved aerosol is infrequent, enormous information is still hidden in the EEM spectra, not to mention the adjacent particle size bins."

15.   Line 83: The use of "But" is not perfect here. The reviewer suggests to use "and" instead.
**Thank you very much for your valuable advice.** Because we have reconstructed and modified the introduction section, this sentence is deleted now.

16.   Line 94: confusing sentence "All samples were collected by quartz filters (Whatman) were prebaked for 5 hours (500°C) and wrapped by aluminum foil stored at -20°C after sampling." May be revised to "All samples collected on quartz filters (Whatman), prebaked for 5 hours (500°C) before sample collection, were wrapped by aluminum foil after sampling and stored at -20°C."
**Thank you very much for your valuable advice.** We have corrected the sentence as suggested. It is now shown as follows.
Line 82-83 "All samples collected on quartz filters (Whatman), prebaked for 5 hours (500°C) before sample collection, were wrapped by aluminum foil after sampling and stored at -20°C."

17.   Line 95: Need clarification? Total 20 groups for 2 seasons or 20 groups each for every season?
**Thank you very much for your valuable advice.** We have added some descriptions of the group sets in line 84, they are now shown as follows.
"A total of 20 groups of 6 stage size segregated aerosol samples were collected at a rural site in Huairou Distinct, Beijing, from 14 November to 30 December 2016, and 30 June to 8 September

2017 for two seasons"

18. Line 106: Should be "The extract was then filtered through a 0.22 μm membrane filter to remove impurities."

**Thank you very much for your valuable advice.** We have corrected the sentence as suggested. It is now shown as follows.

Line 96 "The extracts were then filtered through a 0.22 μm membrane filter to remove impurities (Xiang et al., 2017)."

19. Line 113: Confusing? The sentence may be written like "The extraction procedure of samples subjected to fluorescence and ultraviolet-visible (UV-Vis) measurements were same as WSOC detection."

**Thank you very much for your valuable advice.** We have corrected the sentence as suggested. It is now shown as follows.

Lines 102-103 "The extraction procedures of samples subjected to fluorescence and ultraviolet-visible (UV-Vis) sampling were the same as WSOC detection."

20. Line 117: Should be "Raman Unit"

**Thank you very much for your valuable advice.** We have capitalized "R" in the sentence, they are now shown as follows.

Line 108 "All EEM data in the present research were in the Raman unit (R.U.)"

21. Line 124: Revise the sentence "The EEM data were spectrally corrected by blank sample for instrument bias, inner filter effects, Rayleigh scattering, and most of Raman scatter had been removed" to "The EEM data were spectrally corrected by blank sample to remove interferences from instrument bias, inner filter effects, Rayleigh scattering, and Raman scatter."

**Thank you very much for your valuable advice.** Because we also have been asked to add some explanations of the data correction procedure, by considering both two suggestions, we have corrected the sentences as follows.

Lines 119-123 "All EEM data in the present research were in Raman unit (R.U.), the background signals, interfering signals (first- and second-order Rayleigh and Raman scatterings), and the inner-filter effect were removed by subtracting an EEM of blank, replace with a band of missing values or inserting zeros outside the data area, detailed procedures could be found in Bahram et al., (2006). Data correction and standardization followed procedures described in Xiao et al., (2016)."

22. Line 133-134: Equations number is not matching? Example: "equation (3)" should be "equation (2)" and "equation (4)" should be "equation (3)"

**Thank you very much for your valuable advice.** We are sorry for the unmatching equation number, they are now corrected properly as follows.

$$SS = \frac{1}{\lambda_{Ex}} - \frac{1}{\lambda_{Em}} \qquad\qquad (2)$$

$$WH = 2(\frac{1}{\lambda_{Ex}} + \frac{1}{\lambda_{Em}})^{-1} \qquad\qquad (3)$$

23.   Line 218: Revise "On a large scale of a π-conjugated system, the...."

**Thank you very much for your valuable advice.** We have corrected the phrase as follows.

Line 219 "On a large π-conjugated system, the electron in the ground state needs relatively low excitation energy jumping to the excited state"

24.   Line 222: "Supporting information Figure 3, and Figure 5(c)." shoud be Figure S3 and Figure S5(c). Do same thing for Figures S1, S2, S4 and Table S1, in Supporting Information.

**Thank you very much for your valuable advice.** We Have corrected the citation of supporting information accordingly as Figure S# or Table S#.

25.   Line 87 and 228: The full form of PARAFAC is already mentioned on Line 87. No need to repeat it again. Follow same comment for others as well (e.g. GRA on line 249 etc.).

**Thank you very much for your valuable advice.** We have deleted the full forms of "PARAFAC" and "GRA" after firstly mentioned them in lines 20 and line 138.

---

## Author Response (AR1)

**Dear editor:**

Thank you very much for your consideration of our research "Particle size-dependent fluorescence properties of water-soluble organic compounds (WSOC) and their atmospheric implications on the aging of WSOC". We have carefully addressed the comments of all reviewers, and changed them in the manuscript and attached the corresponding changes after each response. They are listed as following paragraphs, the corrections are started with new line numbers in blue. We have modified the language though out the article by a native English speaker as well. Thank you again for your kindness.

**Reviewer #1**

**For Specific Comments:**

**1. Lines 124-127:**
The text in lines 124-125 is repeating the same information as that provided in lines 115-120. Furthermore, the sentence in lines 126-127 makes more sense in a Introduction section, rather than in a Data analysis section.
**Thank you for your advice.** We have deleted the repeated description in lines 124-125, and moved the sentence in lines 126-127 to lines 53-55 of the Introduction section, they are now read as:
Lines 53-55: "Fluorescence indices are important subsidiary approaches to statistically analyze EEM data (Qin et al., 2018; Yue et al., 2019), which are determined by the chemical structure of pollutants (Andrade-Eiroa et al., 2013a)."

2. **Section 2.4.3.**
Grey relational analysis (GRA): The authors make a strong focus on the novelty of GRA applied to the analysis of EEM fluorescence data. Nevertheless, this is the most obscure section of this study, particularly to those potential readers not familiarized with this analytical tool. The authors should provide a thorough explanation regarding the meaning of each variable in equations (5) and (6) and their relation to the EEM fluorescence data. Furthermore, it is unclear to which factors are the authors referring to when stating that the "fluorescence intensity is highly affected by WSOC concentrations and many other factors (…)" and that "their relations are not clear". Please, be more clear regarding these issues, and explain why you are considering the particles < 0.26 μm as "the references" (only mentioned in line 256)? Moreover, when referring to the "references", do you mean the EEM fluorescence data of WSOC from particles < 0.26 μm? All these issues need to be adequately addressed and thoroughly explained in the manuscript.

**Thank you for your advice.** We are really sorry for so much confusion in the Grey relational analysis (GRA) section. After serious consideration, we have reconstructed this section, left detailed information on reference and comparison sequences of GRA in the manuscript and moved the definition of the GRA method to supplementary information section 1. They are now read as follows.

In manuscript 2.4.3, lines 136-145
"Grey relational analysis (GRA) is part of the grey system theory proposed by Deng (1982), which can be used to describe the relative changes among factors in a system development process. The

detailed calculation of grey relational degree (GRD) was explained in the supplementary information. Generally, in GRA, a reference line and one or a series of comparison sequences were selected, and GRD between the reference line and comparison line indicated the compactness degree. The fluorescence properties of WSOC were considered as a grey system. Two sets of GRA were performed for the WSOC of each season. Firstly, considering the evolution of particle size as a changing system, larger particles might come from accumulation and transformation of smaller particles, especially for ultrafine particles. By setting data of particles smaller than 0.26 μm (WSOC concentrations, AFI or UV) as references and particles larger than 0.26 μm as comparisons, their affinities were analyzed by GRA. Secondly, the fluorescence spectra were generated by part of WSOC, setting the WSOC concentration as a reference and AFI (or UV) as a comparison. The relations between WSOC and AFI for six-stage particles were analyzed."

In SI section 1
"The fluorescence properties present in an EEM spectrum are multifactor triggered results of concentrations, chemical compositions, and even co-existing ions of carbonate species in WSOC. The principle of GRA is to estimate the similarity and degree of the compactness among factors based on the geometric shape of the different sequences. To perform GRA, references and comparison sequences should be selected and converted to the dimensionless format. The grey relational coefficients ξ of the series and grey relational degree are calculated as follows:

$$\xi_i(k) = \frac{\min_i \min_k |y(k)-x_i(k)| + \rho \max_i \max_k |y(k)-x_i(k)|}{|y(k)-x_i(k)| + \rho \max_i \max_k |y(k)-x_i(k)|} \tag{1}$$

$$GRD_i = \frac{1}{n}\sum_{k=1}^{n}\xi_i(k), \ k = 1,2,\dots,n \tag{2}$$

In which $y$ is the reference sequence and $x_i$ ($i$=1,2,3...) is the comparison sequences, $\rho$ is the distinguishing coefficient always set as 0.5, ξ the grey relational coefficients of individual sample of the series, and $GRD_i$ is the grey relational degree calculated by the average of $\xi_i$ (Qiu et al. 2012)."

**3. Lines 166-167:**
The authors state that other researchers also verified a bimodal distribution for the organic matter in other locations within the same region of this study. Firstly, it would be important to clarify whether this bimodal distribution followed the same size distribution as that reported in this study for the summer samples. Secondly, the authors should be aware that the concept of "organic matter" is different from the concept of "WSOC", because in the former you must consider the contribution of atoms (e.g., H, N, S and O) other than carbon to this fraction. This is why it is common to use an aerosol organic mass-to-organic carbon ratio (OM/ OC) to assess the content of organic matter in the air particles in order to achieve a mass closure. Even though it was not possible to estimate the OM/OC ratio in this specific study, it would be interesting to assess whether the organic matter also follows a similar bimodal distribution (see, for example, the work of Li et al. (2020), Science of The Total Environment, 703, 134937, https://doi.org/10.1016/j.scitotenv.2019.134937, for OM/OC ratios for primary and secondary organic aerosols).

**Thank you for your advice.** We are sorry for the neglect here. Firstly, we have added the peaks of recorded bimodal distributions in former research. Secondly, after carefully reading the differences between "WSOC" and "organic matter", we realize that it is inappropriate to use "organic matter"

in line 167, and replace it with "WSOC", besides in the reference articles they are size distributions of WSOC, as well. The sentence is now read as,

Lines 152-153: "Contemporary reports by other researchers observed bimodal distribution of WSOC with two peaks located at 0.8 µm and 7µm, respectively, in Shenzhen, China, and 0.4-0.5 µm and 2-3 µm in Gwangju, Korea (Yu et al., 2016; Huang et al., 2020)."

**4. Line 168:**
Is it possible to include some explanations for the fact that the WSOC/OC ratios are higher in winter than in summer. Could this difference be associated to the prevalence of biomass burning emissions in winter?

**Thank you for your advice.** We add some comparisons of the WSOC/OC in peer works and find that it is hard to conclude why are the WSOC/OC ratios higher in winter than that in summer. but research on the seasonal variations of WSOC in Georgia, US report that in the rural sites non-biomass burning WSOC/OC ratios are higher in winter than that in summer, which is accordant with our present research.

Lines 154-160: "The WSOC/OC ratios were 0.24 - 0.56 in winter and 0.16 - 0.31 in summer. These values were smaller than those previously reported for a polluted period in Beijing and those in the other cities in China (Tian et al., 2014; Wu et al., 2020). Earlier studies suggested higher WSOC/OC ratios in summer than winter (Xiang et al., 2017; Qin et al., 2018), which is in contrast to the results of the present study. Contrasting seasonal patterns in WSOC/OC ratios were also reported between urban and rural sites in Georgia, US (Zhang et al., 2012), which seemed to support our results presented above. The WSOC/OC ratios were higher in particles with an aerodynamic diameter smaller than 1.4 µm than in coarse mode ($PM_{2.5-10}$), which was accordant with findings previously reported for clear days in Beijing (Tian et al., 2016)."

**5. Lines 175-176:**
The authors state that "The bulk fluorescence features of WSOC showed evident distinctions among fine particles and coarse mode particles on EEM spectra". In this Reviewer's opinion, these differences between the EEM spectra of fine and coarse mode particles are more evident in terms of the fluorescence intensity rather than in terms of different fluorescence peaks.

**Thank you for your advice.** We agree with the reviewer's opinion that the fluorescence intensities exhibit more distinctions among fine particles and coarse mode particles than the fluorescence peaks, when firstly looking at the spectra. We tended to express that coarse mode particles had their characteristic EEM spectra, and it seems to confuse. So we discussed them in the later paragraph by FRI and modified the statements in lines 175-176, they are now reading as follows,

Lines 163-168: "The overall fluorescence peaks of EEM were mainly produced among regions II-V and the peaks were peak A, peak T, and peak M, which could be categorized as humic-like, tyrosine-like, and oxygenated organic substances, respectively (Qin et al., 2018). The fluorophores first increased with increasing particle size and reached the highest intensities at particle sizes of 0.26-0.44 µm, and then decreased with increasing particle size in both seasons. Although the

fluorescence peaks of WSOC were mainly produced at similar regions between the two seasons, the relative abundance was different (more quantitative analysis below)."

**6. Lines 197-199:**
The authors state that "FRI â…¢ and FRI â…¤ (HULIS) were the most abundant two fluorophores rich in fine particles." The authors are considering the total fluorescence intensity of these two regions? Figure 4 suggests that FRII is the most abundant fluorophore in fine particles for both summer and winter samples.

Furthermore, the authors also state that "FRI â…£ (microbial related species) peaked between 1.4 to 2.5 μm and showed little variations with particle size increase." However, Figure 4 depicts different results: for the winter samples, FRIV accounts for 23% for particles between 0.26 and 2.5 μm, whereas for the summer samples, the lowest percentage of FRIV (15%) is reported for particles between 1.4 to 2.5 μm. The authors should correct these inconsistencies in their assessment of the results.

**Thank you for your advice.** We are sorry for the inconsistencies in lines 197-199. We have checked the FRI results in Figure 4 and rephrased the description, they are now showed in following sentences.

Lines 184-190: "FRI I and FRI II (protein-like species) increased with increasing particle size and peaked at coarse mode in winter. FRI III and FRI V (HULIS) were mainly abundant in fine particles. FRI IV (microbial related species) showed little variations in particle size range of 0.26-2.5 µm, but decreased with particle size from 2.5 to 10 µm. In summer, the sum of FRI I to FRI III increased with particle size increasing, peaked at 1.4 µm and decreased with particle sizes from 1.4 to 10 µm. FRI IV showed reverse tendencies and decreased with particle size in the range of 0.26 to 1.4 µm, and increase in the particle size range of 1.4 - 10 µm. FRI V didn't have a clear tendency but they showed high portions among 0.26 to 0.44 µm and 0.77 to 1.4 µm."

**7. Lines 238-239:**
If component C3 (assigned to HULIS-2, in line 237) has no physical significance and is considered as a "noise signal", why it is quantified in Figure 6, for the Summer samples? Does it means that 17 to 46% of the fluorescence intensity of PARAFC components for each particle size, in summer samples, is due to "noise signal"? This should be clarified in the manuscript, alongside with a reference to the variance of the model and the core consistency value for each particle size, for the winter and summer samples.

**Thank you for your advice.** The PARAFAC results of summer samples showed an obvious signal of no physical significance. As a fact, we tried 2-7 components PARAFAC analysis, and all results contained this abnormal signal. The spectra were very weak for summer samples, especially for the spectra of large particles at excitation wavelengths between 200-230 nm. We tried to avoid the noise by shrinking the start excitation wavelength to 220 nm, however, the results were barely satisfactory. Thus, an unexpected fraction of 17% to 46% of "noise" was depicted in the results. The 3 components result was selected for its TCC values (Tucker congruence coefficient) of all samples were larger than the threshold of 0.95 and the half-split validation results determined the model was

robust. We added some explanations in the method section of PARAFAC in lines 147-149, they were shown as follows.

Lines 132-134: "Tucker congruence coefficient (TCC) was determined for each excitation spectrum and emission spectrum, and a threshold of 0.95 was applied to confirm the spectral congruence. The model was determined by half-split validation."

**8. Section 3.5:**
As above mentioned, the lack of explanations regarding the GRD analysis applied to the EEM fluorescence data is the main issue of this work. For example, in line 256, which are the comparing factors (and why) and why the particles below 0.26 μm were used as references? In lines 257-258, what do you mean by the statement "The GRD of WSOC, AFI, and UV between particle sizes were basically well among both seasons."?

Furthermore, in lines 267-268, the authors state that "GRD were strongly negatively correlated with estimated secondary organic carbon (SOC) concentrations with correlation efficient r at -0.64 (p<0.000) in winter and -0.63 in summer." Where is the data regarding the estimate of SOC in the collected air particles samples? What was the method followed by the authors to estimate the amount of SOC in the collected samples? Additional data and explanations are required here for a better understanding of how fluorescent WSOC is highly affected by secondary processes, and that GRD between WSOC and AFI could serve as an indicator of secondary formation.

**Thank you for your advice.** Sorry again for the unclear descriptions of the GRA results. As mentioned in question 2, we add the detailed information on the reference and comparison lines of GRA in section 2.4.3 and explains the definition of GRD in SI Section1. Besides, we modified the sentences in section 3.5 and deleted the statements in lines 257-258. Moreover, the data acquirement of secondary organic carbon (SOC) was explained in supplementary information in section 2. We noticed that the GRD of WSOC&AFI shows reversing tendency of decrease first and increase afterwards with minimum values between 0.26 to 0.44 μm, and considering the active accumulation properties of ultrafine particles, we conjectured that secondary processes might affect the fluorescence properties of WSOC, by implementing correlation analysis between GRD and SOC concentrations, strong negative correlation results were found both in winter and summer. We add these thought processes in the manuscript. They are now read as follows.

In 2.4.3 lines 136-145:
"Grey relational analysis (GRA) is part of the grey system theory proposed by Deng (1982), which can be used to describe the relative changes among factors in a system development process. The detailed calculation of grey relational degree (GRD) was explained in the supplementary information. Generally, in GRA, a reference line and one or a series of comparison sequences were selected, and GRD between the reference line and comparison line indicated the compactness degree. The fluorescence properties of WSOC were considered as a grey system. Two sets of GRA were performed for the WSOC of each season. Firstly, considering the evolution of particle size as a changing system, larger particles might come from accumulation and transformation of smaller particles, especially for ultrafine particles. By setting data of particles smaller than 0.26 μm (WSOC concentrations, AFI or UV) as references and particles larger than 0.26 μm as comparisons, their

affinities were analyzed by GRA. Secondly, the fluorescence spectra were generated by part of WSOC, setting the WSOC concentration as a reference and AFI (or UV) as a comparison. The relations between WSOC and AFI for six stage particles were analyzed."

In SI section 1

"The fluorescence properties present in an EEM spectrum are multifactor triggered results of concentrations, chemical compositions, and even co-existing ions of carbonate species in WSOC. The principle of GRA is to estimate the similarity and degree of the compactness among factors based on the geometric shape of the different sequences. To perform GRA, references and comparison sequences should be selected and converted to the dimensionless format. The grey relational coefficients ξ of the series and grey relational degree are calculated as follows:

$$\xi_i(k) = \frac{\min_i \min_k |y(k)-x_i(k)| + \rho \max_i \max_k |y(k)-x_i(k)|}{|y(k)-x_i(k)| + \rho \max_i \max_k |y(k)-x_i(k)|} \tag{1}$$

$$GRD_i = \frac{1}{n}\sum_{k=1}^{n} \xi_i(k), \ k = 1,2,...,n \tag{2}$$

In which $y$ is the reference sequence and $x_i$ ($i$=1,2,3...) is the comparison sequences, $\rho$ is the distinguishing coefficient always set as 0.5, ξ the grey relational coefficients of individual sample of the series, and $GRD_i$ is the grey relational degree calculated by the average of $\xi_i$ (Qiu et al. 2012)."

In SI section 2

"The SOC concentrations were calculated by the method proposed by Castro et al. (1999), as follows:
$$SOC = OC - EC \times (OC/EC)_{min} \tag{3}$$
The results were used to examine the connections between grey relational degree and secondary processes."

In section 3.5

Lines 252-255: "By setting WSOC (or AFI and UV) of particles <0.26 µm as references and those of larger particles as comparisons, relations among particle sizes can be depicted by GRD of size-segregated WSOC (or AFI and UV), as shown in Figure 7(a) for winter samples and Figure 7(b) for summer samples."

Lines 259-260: "The relations between WSOC and AFI and average UV (referred to as UV below) of different particles are shown in Figure 7 (c) and (d) for winter and summer, respectively."

Lines 262-264: "However, clear variations of GRD were observed with increasing particle size with contrasting patterns to those of fluorescence indices. Thus, it was speculated that these variations were resulted from secondary transformation of WSOC, as indicated by fluorescence indices."

**For Technical Corrections:**

1. In this Reviewer opinion, the English language needs extensive revision throughout the manuscript in order to improve not only its reading, but also to clarify the structure and discussion of the scientific results and conclusions.

**Thank you for your advice.** We notice that the language needs to improve at present and have polished the English of this manuscript with professional help.

2.    Line 117: where it reads "Roman unit" it should read "Raman unit".

**Thank you for your advice and very sorry for the mistakes.** We have corrected the wrong character of "Raman" throughout the article. They are now read as follows,

Line 104: "All EEM data in the present research were in Raman unit (R.U.)"

Figure 3: "Size distributions of WSOC and AFI in winter and summer. AFI was in Raman unit."

3.    Line 211: where it reads "p-conjected" it should read "p-conjugated".

**Thank you for your advice and very sorry for the mistake.** We have corrected the word "conjected" to "conjugated" in lines 202-204.

Line 202-204: "Stokes shift (SS) is the energy loss of fluorophore relaxation, which might be associated with the π-conjugated system and electron cloud density (Lakowicz, 2006)."

4.    Line 219: The reference "(Valeur and Berberan-Santos, 2012)" is not accurately listed in the reference list (see line 373).

**Thank you for your advice and very sorry for the mistake.** We have checked the reference list and found that the author order is inversed, it is now corrected.

Lines 213-214: "the electron in the ground state needs relatively low excitation energy jumping to the excited state (Berberan-Santos and Valeur, 2012)"

5.    Line 222: In my opinion, "Figure 3" should appear as "Figure S3" in the text, because the authors are referring to Figure S3 of the Support Information. Please, also see my comment below on Figures S1, S2, S4, S5 and Table S1, in Supporting Information.

**Thank you for your advice and very sorry for the neglect.** The captions in the supplementary information file have all been corrected as Figure S# and Table S#, they are quoted as Figure S# and Table S# in the manuscript as well.

6.    Line 271: where it reads "indicter" it should read "indicator".

**Thank you for your advice and very sorry for the mistake.** We have corrected the word in line 271.

Lines 267-268: "GRD between WSOC and AFI could serve as an indicator of secondary formation."

7.    Figure 1 is not mentioned nor discussed in the manuscript, although it is presented at the end as being part of the manuscript.

**Thank you for your advice and very sorry for the mistake.** We are tending to show the sampling site information in Figure 1 and neglected to refer to it, we add some words in method section 2.1

Lines 84-85: "The air quality index weighted 72h backward trajectories during the sampling period are exhibited in Figure 1."

8.    Figure 2: please, clarify which axis corresponds to the Emission and Excitation wavelengths in order to facilitate the analysis of the EEM spectra by the potential reader.

**Thank you for your advice.** We add "Em" in the horizontal axis and "Ex" in the vertical axis, representing emission and excitation, respectively, in Figure 2

9.    Figure 3 caption: where it reads "Roman unit" it should read "Raman unit".

**Thank you again for your advice.** We have corrected the wrong character of "Raman" in the Figure 3 caption. They are now read as follows,

Figure 3: "Size distributions of WSOC and AFI in winter and summer. AFI was in Raman unit."

10.   Table 1: please, include the units of the WSOC and WSIN concentrations (micrograms per cubic meter?). Moreover, in Table's caption, where it reads "standard divisions" it should read "standard deviations".

**Thank you for your advice and we are sorry for missing units in Table 1.** The chemical concentrations are all in units of mg·m$^{-3}$. We corrected the wrong word of deviations in the caption as well.

**Table 1 Size-segregated average WSOC, WSIN concentrations, and their standard deviations.**

| | Species (mg·m⁻³) | <0.26 µm | 0.26-0.44µm | 0.44-0.77µm | 0.77-1.4µm | 1.4-2.5µm | 2.5-10 µm |
|---|---|---|---|---|---|---|---|
| Winter | $Cl^-$ | 0.42±0.25 | 1.36±1.21 | 0.83±0.72 | 1.03±0.98 | 1.19±1.27 | 0.43±0.45 |
| | $NO_3^-$ | 2.08±1.43 | 9.42±8.46 | 5.64±5.61 | 7.37±8.9 | 6.72±9.44 | 1.92±3.28 |
| | $SO_4^{2-}$ | 1.05±0.6 | 4.36±3.87 | 3.21±3.68 | 5.44±9.43 | 4.68±7.03 | 1.18±1.52 |
| | $Na^+$ | 0.12±0.05 | 0.21±0.1 | 0.16±0.08 | 0.2±0.1 | 0.52±0.6 | 0.24±0.25 |
| | $NH_4^+$ | 1.05±0.57 | 2.9±2.15 | 2.05±1.82 | 2.4±2.77 | 1.67±2.18 | 0.44±0.67 |
| | $Mg^{2+}$ | 0.01 | 0.01 | 0.02±0.01 | 0.05±0.04 | 0.18±0.21 | 0.08±0.09 |
| | $Ca^{2+}$ | 0.06±0.01 | 0.11±0.03 | 0.15±0.08 | 0.4±0.25 | 1.67±1.35 | 0.93±0.9 |
| | $K^+$ | 0.08±0.04 | 0.37±0.3 | 0.24±0.24 | 0.25±0.25 | 0.18±0.18 | 0.05±0.06 |
| | OC | 4.49±1.93 | 11.04±7.2 | 5.67±4.49 | 5.45±6.26 | 5.07±3.88 | 3.4±5.17 |
| | EC | 0.38±0.18 | 0.93±0.47 | 0.67±0.43 | 0.72±0.69 | 0.62±0.78 | 1.65±4.37 |
| | WSOC | 1.66±0.7 | 4.73±2.96 | 2.96±2.41 | 3.21±4.33 | 2.31±2.55 | 0.64±0.5 |
| | WSOC/OC | 0.38±0.07 | 0.43±0.07 | 0.56±0.27 | 0.51±0.15 | 0.37±0.14 | 0.24±0.25 |
| Summer | $Cl^-$ | 0.05±0.02 | 0.1±0.04 | 0.07±0.03 | 0.07±0.02 | 0.16±0.1 | 0.11±0.06 |
| | $NO_3^-$ | 0.48±0.44 | 3.5±3.32 | 1.37±1.35 | 1.04±0.86 | 4.76±4.22 | 1.49±1.37 |
| | $SO_4^{2-}$ | 1.63±1.18 | 7.14±6.64 | 2.59±2.42 | 1.28±1.13 | 0.72±0.51 | 0.2±0.12 |
| | $Na^+$ | 0.29±0.08 | 0.37±0.17 | 0.25±0.06 | 0.23±0.06 | 0.27±0.09 | 0.19±0.03 |
| | $NH_4^+$ | 0.79±0.53 | 2.56±1.99 | 1.18±1.02 | 0.63±0.55 | 0.5±0.46 | 0.1±0.08 |
| | $Mg^{2+}$ | 0.01 | 0.01 | 0.01 | 0.02±0.01 | 0.12±0.08 | 0.05±0.03 |
| | $Ca^{2+}$ | 0.05±0.01 | 0.08±0.02 | 0.08±0.03 | 0.16±0.09 | 1.21±0.87 | 0.62±0.49 |
| | $K^+$ | 0.03±0.02 | 0.14±0.11 | 0.05±0.04 | 0.04±0.02 | 0.06±0.02 | 0.02±0.01 |
| | OC | 2.67±0.98 | 3.93±2.22 | 1.39±0.67 | 1.14±0.41 | 3.5±1.21 | 2.22±1.76 |
| | EC | 0.38±0.12 | 0.44±0.16 | 0.2±0.09 | 0.22±0.06 | 0.34±0.22 | 0.5±0.52 |
| | WSOC | 0.67±0.25 | 1.27±0.86 | 0.46±0.31 | 0.33±0.21 | 0.57±0.18 | 0.27±0.18 |
| | WSOC/OC | 0.26±0.08 | 0.3±0.07 | 0.31±0.1 | 0.27±0.1 | 0.17±0.04 | 0.16±0.12 |

11. Please, update the year of the reference Almeida et al. (Environ. Sci. Technol. 54, 1082-1091), since it was published in 2020.

**Thank you for your advice.** We have modified the publishing year of reference Almeida et al. (2020) both in the manuscript and reference list. They are now read as follows.

 "Water-soluble organic compounds (WSOC) comprise 10% to 80% of organic compounds in atmospheric aerosols (Qin et al., 2018; Almeida et al., 2020; Cai et al., 2020)."

Reference list: "Almeida, A. S., Ferreira, R. M. P., Silva, A. M. S., Duarte, A. C., Neves, B. M. and Duarte, R.: Structural features and pro-inflammatory effects of water-soluble organic matter in inhalable fine urban air particles, Environ Sci Technol, 54, 1082–1091, https://doi.org/10.1021/acs.est.9b04596, 2020."

12. **Supporting information:** the organization and cross-reference, in the main text, of the data presented in the Supporting Information needs to be better addressed. For example, there is no reference in the main text to Figures S1, S2, S4, S5 and Table S1. The authors should also clarify the purpose of these figures and table and how these data were obtained and how it is being used to support the main results and discussion presented in the manuscript. In this regard, as an example, in Figure S2 caption, it is unclear to which particles size correspond the EEM spectra in Figure S2(a), as well as to which studies are the authors referring to in Figure S2(b) and S2(c).

**Thank you very much for your advice.** We are sorry for the neglect of not emphasizing the supporting information in the manuscript. The supplementary information has all been quoted in the manuscript now.

 "Some other fluorescence indices are listed in Table S1"

Table S 1 The grey relational degree of WSOC and AFI between six particle sizes.

| | µm | Pearson correlation | | | Grey relational analysis | | |
|---|---|---|---|---|---|---|---|
| | | a | b | c | a | b | c |
| Winter | 0.26 | 1 | 1 | 0.129 | 1.000 | 1.000 | 0.950 |
| | 0.44 | $0.947^{**}$ | 0.429 | 0.020 | 0.883 | 0.824 | 0.871 |
| | 0.77 | $0.787^{**}$ | $0.724^{**}$ | 0.335 | 0.833 | 0.879 | 0.933 |
| | 1.4 | $0.591^{*}$ | 0.399 | $0.596^{*}$ | 0.766 | 0.830 | 0.928 |
| | 2.5 | $0.637^{*}$ | -0.141 | $0.875^{**}$ | 0.771 | 0.779 | 0.974 |
| | 10 | 0.461 | $0.567^{*}$ | $0.664^{*}$ | 0.808 | 0.840 | 0.982 |
| Summer | 0.26 | 1 | 1 | $0.854^{**}$ | 1.000 | 1.000 | 0.930 |
| | 0.44 | $0.990^{**}$ | $0.943^{**}$ | $0.975^{**}$ | 0.656 | 0.700 | 0.853 |
| | 0.77 | $0.956^{**}$ | $0.920^{**}$ | $0.874^{**}$ | 0.612 | 0.720 | 0.929 |
| | 1.4 | $0.946^{**}$ | $0.825^{*}$ | 0.687 | 0.672 | 0.720 | 0.921 |
| | 2.5 | 0.647 | $0.827^{*}$ | 0.225 | 0.645 | 0.750 | 0.922 |
| | 10 | $0.793^{*}$ | 0.635 | $0.739^{*}$ | 0.577 | 0.641 | 0.948 |

 "The aggregated fluorescence spectra of all size-segregated samples resembled the spectra of TSP and $PM_{2.5}$ shown in Figure S1 with some subtle nuance in border shape (Chen et al., 2016a; Qin et al., 2018)."

(a)

[Figure]

Figure S1 EEM spectra of present research and other studies. (a) The integrated EEM spectra of six-stage particles for winter and summer. (b) and (c) were the fluorophores in Chen et al.(2016) and Qin et al. (2018).

**Lines 173-174:** "the spectra of fine particles widely overlapped with that of PM$_{2.5}$ in Figure S2 (matched with anthropogenic sources and secondary sources of our study),"

[Figure]

Figure S2 The EEM spectra of coarse mode particles in region IV were accordant with the fluorescence regions of biogenic sources.

[Figure]

**Line 217:** "Size-segregated SS, average SS, and $\eta_{SS>1.1}$ are shown in Figure S4 and Figure S5."

Figure S3 Stokes shift of fluorescent WSOC in different particle sizes. Hydrophobic fractions tend to have higher intensity in stokes shifts >1.2, possibly as a result of the larger scale of the $\pi$ conjugated system or higher $\pi$-electron density. In contrast, hydrophilic contents (such as polysaccharides) usually have lower aromaticity and, hence, smaller $\pi$-conjugated systems.

[Figure]

Figure S4 Some fluorescence indices that were not presented in our article. (a) and (b) were the size distribution of average fluorescence intensity of SS (purple bars) and $\eta_{SS>1.1}$ (orange lines) for winter and summer, respectively.

**Reviewer#2:**

**Special comments:**

**1. Line 93: It seems unnecessary to have another repeated "were" at the end of this line. Thank you for your advice.**

We checked the article and added "that" before "were" to lead a clause after filters. They are now shown as follows.

Lines 79-80: "All samples were collected on quartz filters (Whatman), which were prebaked for 5 hours (500°C) before sample collection, and were wrapped by aluminum foil and stored at -20°C."

**2. Line 101: It should be "adapted" rather than "adopted".**

**Thank you for your advice.** We are sorry for the mistake and have corrected it in the sentence.

Lines 87-88: "the thermal evolution protocol IMPROVE (Interagency Monitoring of Protected Visual Environments) was adapted."

**3. Line 115 to 116: Are these parameters for EEM sampling? Please make it clear what are they refer to.**

**Thank you for your advice.** We are sorry for the unclear description in lines 115 to 116. They are now modified as follows.

Lines 103-104: "Briefly, the wavelength ranges of EEM were 200-400 nm for excitation and 250-500 nm for emission with 5 nm interval (Qin et al., 2018)."

**4. Line 155: A verb is missing after "ξ"**

**Thank you for your advice.** We are sorry for the carelessness. We have corrected it, and the description of grey relational analysis has moved to supplementary information now.

SI section 1 "ξ is the grey relational coefficients of individual sample of the series"

**5. Line 181-185: The sentences of this paragraph are hard to read because of lacking the main logic board. Try to describe the seasonality and the size distribution of SFI separately, rather than mixing them.**

**Thank you for your advice.** We are sorry for the confusing descriptions in lines 181 to 185. We have rearranged the logic of this paragraph, by draw primary attention to the fluorophores description, and then describe the seasonal similarities and distinctions of EEM, separately. They are now described as follows.

Lines 162-170: "The size segregated EEM spectra of winter and summer WSOC are depicted in Figure 2 (a) and (b), respectively, and their fluorescence intensities per unit WSOC (SFI) are in (c) and (d), respectively. The overall fluorescence peaks of EEM were mainly produced among regions Ⅱ-Ⅴ and the peaks were peak A, peak T, and peak M, which could be categorized as humic-like, tyrosine-like, and oxygenated organic substances, respectively (Qin et al., 2018). The fluorophores first increased with increasing particle size and reached the highest intensities at particle sizes of

0.26-0.44 µm, and then decreased with increasing particle size in both seasons. Although the fluorescence peaks of WSOC were mainly produced at similar regions between the two seasons, the relative abundance was different (more quantitative analysis below). The aggregated fluorescence spectra of all size-segregated samples resembled the spectra of TSP and PM2.5 shown in Figure S1 with some subtle nuance in border shape (Chen et al., 2016a; Qin et al., 2018)."

**6. Line 196-200: Line 196 to line 198 were mainly about the size distributions of FRI â toâ ¤, however, the description of "FRI â ¢ and FRI â ¤ (HULIS) were the most abundant two fluorophores rich in fine particles." Seems incongruent with the context. Moreover, what is the purpose of adding the reference of Huang et al., (2020) found similar size distribution of protein and HULIS by isotopic method at the end of this paragraph?**

**Thank you for your advice.** We notice that the description of FRI was confusing in lines 196 to 200. The reference of Huang et al., (2020) is unnecessary as well. Thus, the paragraph has been reorganized by separately describing FRI tendencies of winter and summer and the reference is deleted. They are now showed as follows.
Lines 184-190: "FRI I and FRI II (protein-like species) increased with increasing particle size and peaked at coarse mode in winter. FRI III and FRI V (HULIS) were mainly abundant in fine particles. FRI IV (microbial related species) showed little variations in particle size range of 0.26-2.5 µm, but decreased with particle size from 2.5 to 10 µm. In summer, the sum of FRI I to FRI III increased with particle size increasing, peaked at 1.4 µm and decreased with particle sizes from 1.4 to 10 µm. FRI IV showed reverse tendencies and decreased with particle size in the range of 0.26 to 1.4 µm, and increase in the particle size range of 1.4 - 10 µm. FRI V didn't have a clear tendency but they showed high portions among 0.26 to 0.44 µm and 0.77 to 1.4 µm."

**7. Line 243-246: Similar to the former issue, the sentences were uncombined with each other. So the intention of each description is confused. Why do you propose a HULIS1/HULIS2 ratio for winter results? If HULIS1 (or 2) implies a different oxidation state of HULIS, the last sentence should be brought forward.**

**Thank you for your advice.** We checked the phrases in the context and find it is confusing in lines 243-246, because of lacking an explanation on the results of HULIS1/HULIS2 ratio. We add extended the sentence at the end of this paragraph. They are now showed as follows.
Lines 241-245: "The ratios of HULIS-1 / HULIS-2 in winter were higher in fine particles with an aerodynamic diameter of 0.44-2.5 µm than in ultrafine particles (<0.26µm) or coarse mode particles. HULIS-2 was likely freshly emitted fluorescent WSOC and HULIS-1 exhibited fluorescent characteristics of oxidized HULIS (Vione et al., 2019). The low HULIS-1 / HULIS-2 ratios in ultrafine and coarse mode particles might be due to abundant sources of freshly emitted WSOC."

**8. Section 3.5: If it is just as my comprehension, the GRD is a factor of reflecting relations between two factors, why does the author use grey relational analysis rather than correlation analysis?**
**Thank you for your question.** The application of GRD in the present research is a result of comparison. We performed both correlation analysis and GRD between WSOC and AFI, but the

results were not significant for correlation, on this occasion, the GRD results were applied. As depicted in SI Table S1 the results showed that AFI (or WSOC) between particle sizes had great correlations, it is understandable because the size-segregated samples were collected simultaneously. However the correlations between AFI and WSOC were not significant for most particles, which were out of the expectation, besides, the AFI and WSOC didn't fit with any distribution curves as well. In fact, the AFI was the quantified fluorescence property of WSOC, so we tried GRD analysis and the results suggested good connections between AFI and WSOC. We have put the comparison in the supplementary information Section 4.

SI: "**Section 4 Comparison of Pearson correlation analysis and grey relational analysis (GRA) results**

In the manuscript file, we performed grey relational analysis to uncover the underlying connections between WSOC and AFI. Since the fluorescence was generated by part of WSOC, it was conjectured that the AFI and WSOC could be present by mathematical method. We tried the correlation analysis firstly in Table S1. The WSOC (and AFI) of particles <0.26 µm significantly correlated with that of larger particles both in winter and summer. However, the relations between WSOC and AFI were not significant, especially in winter, which was out of the expectation. A possible explanation was that the miscellaneous WSOC in different particles sizes might lead to fortuitous fluorescence intensities.

GRA could reflect the fellowship of factors to the reference line. The relations between WSOC and AFI (Table S1 (c) on the right) were strong in both seasons. Besides, the GRD variation patterns of decrease first and then increase for six particle size stages were just in contrast to that of humification factors, $\eta_{WH>320,}$ and other fluorescence indices. GRD was negatively correlated with SOC ($p<0.01$).

Table S1 The grey relational degree of WSOC and AFI between six particle sizes.

|  | µm | Pearson correlation | | | Grey relational analysis | | |
|---|---|---|---|---|---|---|---|
|  |  | a | b | c | a | b | c |
| Winter | 0.26 | 1 | 1 | 0.129 | 1.000 | 1.000 | 0.950 |
|  | 0.44 | 0.947[**] | 0.429 | 0.020 | 0.883 | 0.824 | 0.871 |
|  | 0.77 | 0.787[**] | 0.724[**] | 0.335 | 0.833 | 0.879 | 0.933 |
|  | 1.4 | 0.591[*] | 0.399 | 0.596[*] | 0.766 | 0.830 | 0.928 |
|  | 2.5 | 0.637[*] | -0.141 | 0.875[**] | 0.771 | 0.779 | 0.974 |
|  | 10 | 0.461 | 0.567[*] | 0.664[*] | 0.808 | 0.840 | 0.982 |
| Summer | 0.26 | 1 | 1 | 0.854[**] | 1.000 | 1.000 | 0.930 |
|  | 0.44 | 0.990[**] | 0.943[**] | 0.975[**] | 0.656 | 0.700 | 0.853 |
|  | 0.77 | 0.956[**] | 0.920[**] | 0.874[**] | 0.612 | 0.720 | 0.929 |
|  | 1.4 | 0.946[**] | 0.825[*] | 0.687 | 0.672 | 0.720 | 0.921 |
|  | 2.5 | 0.647 | 0.827[*] | 0.225 | 0.645 | 0.750 | 0.922 |
|  | 10 | 0.793[*] | 0.635 | 0.739[*] | 0.577 | 0.641 | 0.948 |

** Correlation is significant at the 0.01 level.

*   Correlation is significant at the 0.05 level.

a. GRA was performed by setting WSOC of 0.26 µm as references and the rest particle sizes as a comparison.

b. GRA was performed by setting AFI of 0.26 µm as references and the rest of particle sizes as a comparison.

c. GRA was performed by setting WSOC of each particle size as references and corresponding AFI as a comparison."

**9.    The author state that "Our unpublished research found that the AFI/WSOC ratios were lower than 0.2 for anthropogenic source samples, indicating that this ratio might be higher in oxidized fluorescent WSOC." This "indicating" may not be easily deduced here, and I noticed that these inductions are discussed in lines 282 to line 292, so the description in line 191 can be saving for the later paragraph.**

**Thank you for your advice.** We agree with your perspective that the deduction is inconsequence, and we have deleted it in line 191, they are now shown as follows.

Lines181-182: "Our unpublished research found that the AFI/WSOC ratios were lower than 0.2 for anthropogenic source samples."

**10.    Line 293: The first sentence shows weak leadership for this paragraph, it also shows little connections with later context**

**Thank you for your advice.** We have reconstructed the paragraph and sentence. They are now shown as follows.

Lines 273-277: "Accordant with earlier reports, the fluorescence intensities positively correlated with WSOC concentrations in both winter and summer (Spearman's $r$>0.8, $p$<0.001) (Qin et al., 2018; Chen et al., 2019). The size distributions of AFI kept in step with those of WSOC concentrations and showed monomodal distributions in winter and bimodal distributions in summer, peaking in particle sizes between 0.26 to 0.44 µm (Figure 2 (a) and (b)). The EEM spectra of size-segregated WSOC mainly exhibited among regions II-V and blue-shifted with increasing particle size (0.44 to 10 µm)."

**11.    Line 315 to 318: The conjectured sources of HULIS are not closely related to the former context.**

**Thank you for your advice.** We realized the abrupt deduction on sources of WSOC is reasonless, thus the sentence is deleted now. The context is shown as follows.

Lines 313-317: "In winter, the wavelength of HULIS-1 was slightly higher than that of HULIS-2. The EEM spectra of HULIS-1 observed in the present study were similar to the PARAFAC results of highly oxygenated species, while those of HULIS-2 to less oxygenated species reported in Chen et al (2016b) on the chromophoric WSOC. Only HULIS-1 was distinguished in summer in the present study, which could be allocated to highly oxygenated species."

**Minor issues:**

1.The tense form should be unified

**Thank you for your advice.** We have corrected the tense in following lines.

Lines 93: "The extracts **were** then filtered through a 0.22 µm membrane filter to remove impurities

(Xiang et al., 2017)."

Lines 284-285: "The specific fluorescence area was widened in ambient sample and thus had a higher AFI/WSOC ratio when WSOC concentrations **were** at a comparable level."

2.Check the abbreviations and capitalized letters throughout the article, some of them are in the wrong format.

**Thank you for your advice.** We have corrected abbreviations and capitalized letters in the article.

3.Some of the definite or indefinite articles are missing.

**Thank you for your advice.** We have corrected the definite or indefinite articles in the following lines.

Line 92: "A quarter of **the** filter sample was ultrasonically extracted twice with 5 ml ultrapure water each time and mixed up after extraction."

Lines 102-103: "Briefly, **the** wavelength ranges of EEM were 200-400 nm for excitation and 250-500 nm for emission with 5 nm interval (Qin et al., 2018)."

Line 234: "C2 representing **a** protein-like component,"

Line 322: "which confirmed that GRD value can be applied as an indicator of the aging state of WSOC."

**Reviewer #3**

**Major Comments:**

**1. Introduction:**

Lacks organization and continuity? The reviewer suggests the revision of the introduction section to make it more organized and in tandem with the objective of the study.

**Thank you very much for your valuable advice.** We have reorganized the introduction section into four parts: Paragraph one is the general topic of WSOC; paragraphs two and three introduce several advanced analytical methods that have been used in recent research and refers their limitations, then the advantage of optical methods is proposed; paragraph four lists several recent research on size segregated WSOC; and the perspective of the present research is summarized in the end. The new introduction is shown as follows.

Lines 27-74: "The environmental, health, and climate effects of atmospheric aerosol particles have been reiterated for many years (Pósfai and Buseck 2010; Burnett et al., 2018; Yan et al., 2020; Fan et al., 2020). Water-soluble organic compounds (WSOC) comprise 10% to 80% of organic compounds in atmospheric aerosols (Qin et al., 2018; Almeida et al., 2020; Cai et al., 2020). WSOC play significant roles in cloud formation, solar irradiation, and atmospheric chemistry (Asa-Awuku et al., 2009; Duarte et al., 2019). However, only 10% to 20% of the organic compounds have been structurally identified, and the majority of WSOC remain uncharacterized. Generally, WSOC mixture contains both aromatic nuclei and aliphatic chains (Decesari et al., 2001; Dasari et al., 2019), with functional groups or heteroatoms like hydroxyl, carboxyl, aldehyde, ketone, amino, and other nitrogen-containing groups (Duarte et al., 2007; Cai et al., 2020). Biomass burning and secondary transformation of organics are believed to be the main sources of WSOC (Park et al., 2017; Xiang et al., 2017).

[revised manuscript text omitted]

**2. Line 161:**

The size distribution of WSOC/OC and WSOC concentration doesn't follow similar trend. Although several studies in the past (Dasari et al., 2019 science advances; Choudhary et al., 2021 environmental pollution), as well as this study (in introduction), have stated that majority of WSOC are secondary (oxidized) in nature. The author can elaborate possible rationales briefly?

**Thank you very much for the question.** We are sorry for not discussing the size distribution of WSOC/OC ratios and WSOC. As a result, the size distribution of WSOC/OC and WSOC concentration doesn't follow a similar trend indeed. We can see in Figure A. below, that WSOC and OC show similar tendencies for both seasons, and the peaks of WSOC/OC show a delay (peaks around 1um) comparing to their concentrations. This may be because of an increased portion of WSOC or a decrease of OC, as we all know, WSOC is part of OC, thus, we prefer to believe that

more OC oxidized to WSOC in particle sizes around 1um, since particles of those sizes can long-exist in the atmospheric environment.

Another possible reason might be that the difference in sources and transformation process of size segregated particles might lead to multiple WSOC/OC results. Ram et al. (2012) reported the WSOC/OC ratio of 0.47±0.11, and characteristically listed the former reported results in vehicle exhaust aerosol and biomass burning affected aerosol, and found that the ambient aerosol had a higher WSOC/OC value and concluded that WSOC/OC could serve as an indicator of secondary formation.

The author stated that "The AFI/WSOC ratios ranged from 0.22 to 0.57 in winter and 0.18 to 0.34 in summer, respectively." "Our unpublished research found that the AFI/WSOC ratios were lower than 0.2 for anthropogenic source samples, indicating that this ratio might be higher in oxidized fluorescent WSOC." If that is the case, size distribution of AFI/WSOC should have follow the distribution trend of WSOC/OC (a tracer for photochemical oxidation), but this is not the case in this paper (Figure 3). Explain the rationale/s behind this behaviour?

**Thank you very much for your valuable advice.** We also have noticed this inappropriate deduction during the revision period, thus the indication sentences of "..., indicating that this ratio might be higher in oxidized fluorescent WSOC." have been deleted in line 193. Because the

phenomenon of low AFI/WSOC values in sources samples and PM$_{2.5}$ samples in a polluted city and relatively high values in ambient environment samples cannot strongly confirm this conjecture, and we provide some possible explanations instead in the Discussion section. The different size distributions of AFI/WSOC and WSOC/OC may be because that oxidization of WSOC causes fluorescence quenching, we mentioned this perspective in lines 295-298. We have reconstructed the sentences in lines 186-193 and 285-290 as follows.

Lines 179-182: "AFI/WSOC ratios could represent the overall average fluorescence density of WSOC (Xiao et al., 2016). The AFI/WSOC ratios ranged from 0.22 to 0.57 in winter and from 0.18 to 0.34 in summer. These values were higher than that in the industrial city of Lanzhou (Qin et al., 2018). Our unpublished research found that the AFI/WSOC ratios were lower than 0.2 for anthropogenic source samples."

Lines 282-287: "Substitution and oxidation reactions of ambient organics might widen the delocalization of π electronics and reduce the excitation energy, thereby resulting in a redshift of EEM spectra (Kalberer et al., 2004). The specific fluorescence area was widened in ambient sample and thus had a higher AFI/WSOC ratio when WSOC concentrations were at a comparable level. Continuous oxidation of organics may break up the π system of organics and extinct fluorescence (Zanca et al., 2017). It could be inferred that ambient WSOC tended to exhibit higher AFI/WSOC ratios, while both freshly emitted WSOC and completely oxidized WSOC could lead to lower AFI/WSOC values."

**5. Line 212:**

The author stated that Stokes shift (SS) of 1.2 μm-1 is an important border of hydrophobic and hydrophilic components. And later used Stokes shift of 1.1 to determine ratios of fluorescence intensity in high SS. Elaborate the possible reason/s?

**Thank you very much for the question.** We are sorry for not explaining the reason for using SS of 1.1um$^{-1}$ as the border of hydrophobic and hydrophilic components. As a fact, Xiao's former research found that hydrophobic fractions tended to present fluorescence peaks at SS>1 um$^{-1}$, thus we use the average value of SS at 1.1 um$^{-1}$ as the border, we add the description in line 224. The context is now shown as follows.

Lines 205-210: "Xiao et al., (2019) found that SS near 1.2 μm$^{-1}$ is an important border of hydrophobic and hydrophilic components. Hydrophobic fractions tend to have higher intensity in SS >1.2, possibly as a result of the large scale of the π conjugated system. In contrast, hydrophilic fractions usually have ionogenic groups bond with fluorescent aromatics reduced π-conjugated systems, hence, leading to high fluorescence intensities existing on both sides of SS of 1.2. Note that the same research also reported earlier that hydrophobic fractions tended to present fluorescence peaks at SS >1 (Xiao et al., 2016). Thus, the ratios of fluorescence intensity in high SS (>1.1) are calculated as follows:"

**6. Line 205:**

HIX (aromaticity) and WSOC/OC (oxidation) ratio following same size distribution trend. How come? This could be an important finding of the manuscript. Add some discussion about same in Discussion and Implication sections.

**Thank you very much for your valuable advice.** We have compared the size distributions of HIX

and WSOC/OC ratio, find that they both show monomodal distributions and the peaks of WSOC/OC delay to larger particle sizes comparing to HIX. That means the HIX starts to decrease when WSOC/OC still increases. The decrease of HIX maybe because the fluorescent WSOC is oxidized to non-fluorescent organics during a long period of exposure in the ambient environment, as the fluorescence property requires conjugated systems in organics. While the long oxidation period also leads to more OC convert to WSOC at the same time, thus the WSOC/OC ratio keeps increase. We have added some discussion in lines 304-307 as follows.

Lines294-297: "Besides, it was noticed that HIX and WSOC/OC showed similar size distributions except for larger peaking particle size of WSOC/OC values comparing to HIX. Because the fine particles with relatively large sizes could long exist in atmospheric environment, the WSOC/OC ratios increase gradually, however, the oxidation process could also cause fluorescence quenching and lead to the decrease of HIX (Vione et al., 2019). Thus, HIX peaked in smaller particle size comparing to WSOC/OC."

**7.  Line 209, 210, 243:**
The author categorized protein-like compounds into biogenic origin. But aerosols partitioned from VOCs (isoprene etc.) emitted from plants also categorized into biogenic aerosols. Does the author also incorporating these aerosols produced from VOCs in Protein-like compounds or it is just bioaerosols? Please clarify?

**Thank you very much for your valuable advice.** We are sorry for the unclear description of biogenic sources. In the present research, we have only considered protein-like compounds in the particulate organics as biogenic and neglected that biogenic VOC are biogenic aerosols as well. To avoid misunderstanding, we have changed the "biogenic sources" to "microbial related" throughout the article. The "biogenic oxygenated organics" in line 215 is a term of Huang's research, thus we keep it unchanged. The other corrections are as follows.
Line 197-198: "Peak T/Peak C peaked at coarse mode in both seasons, indicating that fluorescent microbial related species likely existed in large atmospheric particles."
Line 241: "demonstrated that microbial related WSOC were more likely to exist in large particles"

**8.  Line 256:**
Why did author used particles <0.26 µm as references for Grey relational analysis (GRA)? Why not use size bins where WSOC, UV and AFI are maximum?

**Thank you very much for the question.** We are sorry for not carefully explain the reasons for reference list selection. The minimum particle size is selected because it is assumed that the increase of particle size is an accumulation process, we are trying to find the connections between different particle sizes during the particle increase processes. We have added some explanations in Section 2.4.3, they are shown as follows.
Lines 141-145: "Firstly, considering the evolution of particle size as a changing system, larger particles might come from accumulation and transformation of smaller particles, especially for ultrafine particles. By setting data of particles smaller than 0.26 µm (WSOC concentrations, AFI or UV) as references and particles larger than 0.26 µm as comparisons, their affinities were analyzed by GRA."

**Minor Corrections:**

In the Reviewer's opinion, the English language needs significant revision throughout the manuscript before acceptance. The some of English-related corrections and other minor comments are suggested below:

**We are sorry for the mistakes in the last version of the manuscript.** We have carefully addressed all of the suggestions and corrected them, we also have checked through the article and modified them. The language has been polished by professionals.

1. Line 33: Replace "mysterious" with either "Unknown" or "Uncharacterized".

**Thank you very much for your valuable advice.** We have changed "mysterious" to "Uncharacterized" in lines 31-32, they are now shown as follows.

Lines 31-32: "However, only 10% to 20% of the organic compounds have been structurally identified, and the majority of WSOC remain uncharacterized."

2. Line 34-35: The sentence lacks continuity. Revise the sentence "incorporating with different highly oxidized functional groups or heteroatoms like" with may be something like "WSOC mixture contains both aromatic nuclei and aliphatic chains (Decesari et al., 2001; Dasari et al., 2019), with functional groups or heteroatoms like hydroxyl, carboxyl, aldehyde, ketone, amino, and other nitrogencontaining groups (Duarte et al., 2007; Cai et al., 2020)".

**Thank you very much for your valuable advice.** We have revised this sentence as follows.

Line 32-34: "Generally, WSOC mixture contains both aromatic nuclei and aliphatic chains (Decesari et al., 2001; Dasari et al., 2019), with functional groups or heteroatoms like hydroxyl, carboxyl, aldehyde, ketone, amino, and other nitrogen-containing groups (Duarte et al., 2007; Cai et al., 2020)."

3. Line 37: Is the reference "(ParkSeungShik et al., 2017)" is correctly cited and listed in the reference list (also see line 485).

**Thank you very much for your valuable advice.** We have checked the reference list and modified the citation format in the text. The citations are shown as follows.

In the text line 35: "(Park et al., 2017)"

In the reference list "Park S., Yu, J., Yu, G.-H. and Bae M. S.: Chemical and absorption characteristics of water-soluble organic carbon and humic-like substances in size segregated particles from biomass burning emissions, Asian J. Atmos. Environ., 11, 96-106, https://doi.org/10.5572/ajae.2017.11.2.096, 2017."

4. Line 39: Revise "Nuclear magnetic resonance (NMR) and mass spectrometry (MS) are two remarkable analytical methods using to structurally unravel the complex WSOC (Duarte et al., 2020)."

**Thank you very much for your valuable advice.** We have reconsidered the context and modified the whole paragraph. This sentence is deleted now.

5. line 46: It is "Accelerator" not "accelerate".

**Thank you very much for your valuable advice.** Because we have reconstructed and modified

the introduction section, this sentence is deleted now.

6.  Line 46: Revise the sentence to something like "Isotopic ratio mass spectroscopy (IRMS) and accelerator mass spectroscopy (AMS) are widely used to distinguish organic emissions from fossil combustion sources and biogenic sources using carbon isotopic characteristics (Masalaite et al., 2018; Zhao et al., 2019; Huang et al., 2020)."

**Thank you very much for your valuable advice.** We have corrected the sentence, they are now shown as follows.

lines 38-43: "Applications of other existing technologies used for identifying organics structure include the electrospray ionization with ultrahigh-resolution Fourier-transform ion cyclotron resonance mass spectrometry (ESI-FT-ICR-MS), the proton transfer reaction mass spectrometry (PTR-MS), the Isotopic ratio mass spectroscopy (IRMS), and the accelerator mass spectroscopy (AMS), have been increasing because the requirement of further insight into organics in particulate matter (Cai et al., 2020; Mayorga et al., 2021), and suource distinguishment of organic emissions from fossil combustion or biogenic origin (Masalaite et al., 2018; Zhao et al., 2019; Huang et al., 2020)."

7.  Line 50-56: Whole paragraph lacks organization and continuity. The reviewer suggests the revision of the paragraph.

**Thank you very much for your valuable advice.** We have reconstructed the paragraph as follows.

Lines 44-55: "The above-mentioned instruments are generally expensive to operate. In contrast, optical instruments like ultraviolet and fluorescence spectrophotometers are relatively low-cost and efficient. Moreover, data generated by the optical instruments can provide quantitative and qualitative information simultaneously, which warrants their broad application on organics research, such as investigating WSOC and dissolved organic matter (DOM) in water (Hecobian et al., 2010; Qin et al., 2018; Xiao et al., 2016). 3-Dimensional excitation-emission matrix (EEM) fluorescence spectroscopy is an optical instrument that has been used in analyzing atmospheric WSOC (Duarte et al., 2004; Fu et al., 2014). Fluorescence analysis can identify chromophoric organics like aromatics, protein, and other organic matters containing π-conjugated systems (Xiao et al., 2018; 2020). EEM spectrum has been implemented to visualize the fluorescence regions and identify possible categories of WSOC by characteristic of fluorescent regions (Duarte et al., 2004; Santos et al., 2009), and to study the aging of WSOC by examining the red or blue shift of fluorescence peaks (Lee et al., 2013; Fu et al., 2015; Vione et al., 2019). Fluorescence indices are important subsidiary approaches to statistically analyze EEM data (Qin et al., 2018; Yue et al., 2019), which are determined by the chemical structure of pollutants (Andrade-Eiroa et al., 2013a)."

8.  Line 57: Replace "3-Dimensional fluorescence of excitation-emission matrix (EEM)" to "3-Dimensional excitation–emission matrix (EEM) fluorescence spectroscopy"

**Thank you very much for your valuable advice.** We have corrected the phrase as suggested.

Lines 48-49: "3-Dimensional excitation-emission matrix (EEM) fluorescence spectroscopy is an optical instrument that has been used in analyzing atmospheric WSOC (Duarte et al., 2004; Fu et al., 2014)."

9. Line 59: it should be "mainly helpful in investigating"

**Thank you very much for your valuable advice.** This sentence has been deleted in the revision processes.

10. Line 62: what does author mean by "in early years"? Does author mean "earlier studies", if so, revise the sentense.

**Thank you very much for your valuable advice.** we have modified the phrase in lines 52-54 as follows.

Lines 50-52: "EEM spectrum has been implemented to visualize the fluorescence regions and identify possible categories of WSOC by characteristic of fluorescent regions (Duarte et al., 2004; Santos et al., 2009),"

11. Line 65: It should be "analyse" not "analysis"

**Thank you very much for your valuable advice.** Because we have reconstructed and modified the introduction section, this sentence is deleted now.

12. Line 69: "(great parts of WSOC)"? It should be something like "significant fraction of WSOC"

**Thank you very much for your valuable advice.** We have corrected the phrase accordingly as follows.

Lines 59-60: "Structural investigations on coal burning and biomass burning affected humic-like substances (a significant fraction of WSOC) in four size ranges found consistent organic specie through all the size ranges,"

13. Line 70: "reversely"?

**Thank you very much for your valuable advice.** We have changed "but" to "however" in line 60, they are now shown as follows.

Lines 60-61: "however, the absorption bands of aromatic groups were more intense compared to carboxylic groups in sub-3 µm fractions (Park et al., 2017; Voliotis et al., 2017)."

14. Line 82: "neighbor particle sizes" should "adjacent particle size bins"

**Thank you very much for your valuable advice.** We have reconstructed the sentence.

Lines 68-69: "To date, comprehensive analysis of fluorescence properties of size-resolved aerosols is still very limited, with enormous information being hidden in the EEM spectra."

15. Line 83: The use of "But" is not perfect here. The reviewer suggests to use "and" instead.

**Thank you very much for your valuable advice.** Because we have reconstructed and modified the introduction section, this sentence is deleted now.

16. Line 94: confusing sentence "All samples were collected by quartz filters (Whatman) were prebaked for 5 hours (500°C) and wrapped by aluminum foil stored at -20°C after sampling." May be revised to "All samples collected on quartz filters (Whatman), prebaked for 5 hours (500°C) before sample collection, were wrapped by aluminum foil after sampling and stored at -20°C."

**Thank you very much for your valuable advice.** We have corrected the sentence as suggested. It is now shown as follows.

 "All samples were collected on quartz filters (Whatman), which were prebaked for 5 hours (500°C) before sample collection, and were wrapped by aluminum foil and stored at -20°C."

17.     Line 95: Need clarification? Total 20 groups for 2 seasons or 20 groups each for every season?

**Thank you very much for your valuable advice.** We have added some descriptions of the group sets in line 81, they are now shown as follows.

Lines 81-82: "A total of 20 sets of 6-stage size segregated aerosol samples were collected at a rural site in Huairou Distinct, Beijing, from 14 November to 30 December 2016 and from 30 June to 8 September 2017."

18.     Line 106: Should be "The extract was then filtered through a 0.22 μm membrane filter to remove impurities."

**Thank you very much for your valuable advice.** We have corrected the sentence as suggested. It is now shown as follows.

Line 93: "The extracts were then filtered through a 0.22 µm membrane filter to remove impurities (Xiang et al., 2017)."

19.   Line 113: Confusing? The sentence may be written like "The extraction procedure of samples subjected to fluorescence and ultraviolet-visible (UV-Vis) measurements were same as WSOC detection."

**Thank you very much for your valuable advice.** We have corrected the sentence as suggested. It is now shown as follows.

Lines 102-103: "The extraction procedures of samples subject to fluorescence and ultraviolet-visible (UV-Vis) sampling were the same as for WSOC detection."

20.   Line 117: Should be "Raman Unit"

**Thank you very much for your valuable advice.** We have capitalized "R" in the sentence, they are now shown as follows.

Line 104: "All EEM data in the present study were in Raman unit (R.U.)."

21.   Line 124: Revise the sentence "The EEM data were spectrally corrected by blank sample for instrument bias, inner filter effects, Rayleigh scattering, and most of Raman scatter had been removed" to "The EEM data were spectrally corrected by blank sample to remove interferences from instrument bias, inner filter effects, Rayleigh scattering, and Raman scatter."

**Thank you very much for your valuable advice.** Because we also have been asked to add some explanations of the data correction procedure, by considering both two suggestions, we have corrected the sentences as follows.

Lines 104-108: "All EEM data in the present study were in Raman unit (R.U.). The background signals, interfering signals (first- and second-order Rayleigh and Raman scatterings), and the inner-filter effects were removed by subtracting an EEM of blank and replaced with a band of missing values or inserting zeros outside the data area, as detailed in Bahram et al., (2006). Data correction and standardization followed procedures described in Xiao et al., (2016)."

22. Line 133-134: Equations number is not matching? Example: "equation (3)" should be "equation (2)" and "equation (4)" should be "equation (3)"

**Thank you very much for your valuable advice.** We are sorry for the unmatching equation number, they are now corrected properly as follows.

$$SS = \frac{1}{\lambda_{Ex}} - \frac{1}{\lambda_{Em}} \tag{2}$$

$$WH = 2(\frac{1}{\lambda_{Ex}} + \frac{1}{\lambda_{Em}})^{-1} \tag{3}$$

23. Line 218: Revise "On a large scale of a π-conjugated system, the...."

**Thank you very much for your valuable advice.** We have corrected the phrase as follows.

Line 213-214: "On a large π-conjugated system, the electron in the ground state needs relatively low excitation energy jumping to the excited state (Berberan-Santos and Valeur, 2012)."

24. Line 222: "Supporting information Figure 3, and Figure 5(c)." shoud be Figure S3 and Figure S5(c). Do same thing for Figures S1, S2, S4 and Table S1, in Supporting Information.

**Thank you very much for your valuable advice.** We Have corrected the citation of supporting information accordingly as Figure S# or Table S#.

25. Line 87 and 228: The full form of PARAFAC is already mentioned on Line 87. No need to repeat it again. Follow same comment for others as well (e.g. GRA on line 249 etc.).

**Thank you very much for your valuable advice.** We have deleted the full forms of "PARAFAC" and "GRA" after firstly mentioned them in lines 20 and line 138.

---

## Author Response (AR2)

Dear editor Alex Huffman:

Thank you very much for your considerations on our manuscript "Measurement Report: Particle size-dependent fluorescence properties of water-soluble organic compounds (WSOC) and their atmospheric implications on the aging of WSOC", MS No.: acp-2021-465. And thank you for the recognition of our work. We have carefully checked through the manuscript and revised the Figures and vague descriptions as suggested, detailed changes are listed below. A manuscript with tracked changing version and a clean version are uploaded.

Line numbers refer to the submitted "tracked changes" version manuscript.

**1) Please adapt title to begin as "Measurement Report: Particle size-dependent …"**

**Response:** Thank you for sorting the manuscript to the best fit form of "Measurement Report". We have added "Measurement Report:" before the original title, they are now read as "Measurement Report: Particle size-dependent fluorescence properties of water-soluble organic compounds (WSOC) and their atmospheric implications on the aging of WSOC"

**2) Abstract, line 12: The mention of fluorescence spectroscopy having been used to "investigate the sources" is a bit of a stretch. It would be more accurate to say something like "…fluorescence spectroscopy was used to investigate optical properties of WSOC as means of inferring information about their atmospheric sources."**

**Response:** Thank you for your advice. The statement of "fluorescence spectroscopy was used to investigate the sources of WSOC" might be over stretched, so we have change them as advised in the Abstract. They are now shown as follows.

Lines 12-13: "3-Dimensional fluorescence spectroscopy was used to investigate the optical properties of WSOC as means of inferring information about their atmospheric sources."

3) I'm not comfortable with the use of the term "EEM" in all cases throughout the manuscript. An EEM is really just a way of plotting fluorescence emission spectra acquired at many separate excitation wavelengths, but doesn't represent a type of spectroscopy fundamentally different from other kinds of fluorescence spectroscopy. For example, in line 48 you state that EEM fluorescence spectroscopy "is an optical instrument." I would change that to say EEMs can be extracted from fluorescence spectra acquired on a fluorescence spectrometer.

**Response:** Thank you for your advice, and we are sorry for the confusing usage of "EEM" in the manuscript. We have checked through the article and replaced "EEM" with "fluorescence" or deleted them according to the context. Detailed changes with track are listed as follows.

For the inappropriate usage of EEM in line 48, we changed it to "3-Dimensional fluorescence spectroscopy", and added the explanation of EEM in line 51-53.

Line 48-49: "3-Dimensional  fluorescence spectroscopy is an optical instrument that has been used in analyzing atmospheric WSOC"

Line 51-53: "Excitation-emission matrix (EEM) can be extracted from fluorescence spectra (acquired on a fluorescence spectrometer) and visualized to show fluorescence regions and possible categories of WSOC by the spectral characteristics"

Other changes of "EEM" in the manuscript:

Line 11-12: "3-Dimensional fluorescence spectroscopy was used to investigate ..."

Line 13: "Sophisticated data analysis on fluorescence data was performed to..."

Line 16: "The excitation-emission matrix (EEM) spectra of WSOC varied with particle size..."

Line 56: "... are important subsidiary approaches to statistically analyze fluorescence properties of WSOC."

Line 104: "the wavelength ranges  were 200-400 nm for excitation and 250-500 nm for emission with 5 nm interval for fluorescence spectroscopy"

Line 172: "The overall fluorescence peaks  were..."

Line 180: "The detailed characteristics of fluorescence spectra could be found in SFI spectra."

Line 201: "Inclusive information was stored in fluorescence spectra..."

Line 233: "3.4 Fluorophores revealed by classification of PARAFAC results"

Line 242: "...to reveal seasonal dependent fluorescence spectra. Three components were extracted from winter  spectra, including..."

Line 292: "...thereby resulting in a redshift of fluorescence spectra..."

Line307: "All evidence on fluorescence spectra and  indices discussed above suggested..."

4) Thank you for your edits to Section 2.4.3, however I still find the section somewhat confusing. The definitions of the term GRA and GRD are somewhat confusing, in part because of typos. Section 2.4.3 lists these two definitions: "grey relational analysis (GRA)" and "grey relational degree (GRD)". But then Section 3.5 and again in Section 1 (line 106) re-defines GRD as "Gary Relational Degree." Please make these consistent and only define each acronym a total of one time.

Please focus one more time to clarify the development of this topic. The explanation of this section should be understandable without need to reference the supplemental section discussing GRD. As examples:

- In context of this manuscript, it isn't clear what a 'system development process' is (line 179).

- The first sentence introduces GRA, and then the second sentence says that GRD can be calculated using information in the supplement.

- Line 185: "A reference line" – Does this mean a row or column of the EEM spectral matrix? What sequences do you mean? Other spectra?

- What defines a "grey system?" (line 178, 189). Please clarify in context of this manuscript.

**Response:** Thank you for your advice, and we have reconstructed section 2.4.3 into two parts. In the first paragraph, we explained grey relational analysis, grey system and the applicability of grey relational analysis in environmental research; and then the relations of grey relational analysis and grey relational degree (GRD), the calculation criterion of GRD was explained in brief. While, in the second paragraph, the GRD calculated in the present research was explained.

We also have corrected the confusion statement accordingly. Firstly, the abbreviation of GRA is replaced with "grey relational analysis" and only GRD left to avoid confusion. Secondly, "a system development process" was trying to refer to "grey system" which were not defined in the manuscript. Thus, we have deleted the words "development process" and added some descriptions of grey system. Lastly, the reference line and comparing line are explained in the second paragraph of section 2.4.3.

The revised section 2.4.3 are shown as follows:

[revised manuscript text omitted]

6) Figures:

a. Figure 2: Clarify what "Ex" and "Em" are in the caption, making clear these are wavelengths of excitation or emission, and including the units in the caption. Please also clarify why the x-axis legend only covers one of the six columns.

**Response:** We are sorry for missing annotative information in the graph, we have modified them now.

[Figure]

**Figure 2 EEM spectra of size segregated samples in winter and summer, their excitation and emission wavelength range were the same and only showed in first EEM of (d). All spectra were partitioned into five regions and assigned as protein-like pollutants (I and II), fulvic acid (III), soluble microbial byproduct-like substances (IV), and humic-like acid (V), respectively (Birdwell and Engel 2010). Peak A, B, C, M, and T were generally considered as humic-like fluorophores, tyrosine-like fluorophores, humic-like carbon with larger molecular weight, marine humic-like fluorophore, and tryptophan-like fluorophores (Coble 1996). (a) and (b) were the size segregated EEM spectra of winter and summer samples, respectively (Unit: R.U.), (c) and (d) were the corresponding EEM spectra of fluorescence emitted per unit of WSOC carbon (Unit: R.U.·L·mg⁻¹).**

b. Figure 4: Please place Fluorescence regional integration (FRI) somewhere in the figure caption. Additionally, the values on the x-axis run into one another. Don't make the font size any smaller, but otherwise consider how to revise the x-axis. No y-axis label.

Response: **Thank you for your advice,** we have added "Fluorescence regional integration (FRI)", rotated the x-axis label and added y-axis label in Figure 4.

[Figure]

**Figure 4 Size distribution of fluorescence regional intensity for winter and summer. FRI1-FRI5 was FRI of fluorescence region I to V.**

c. Figure 6: The font size on the axes and legends of this figure is too small to read easily.

**Response: Thank you for your advice,** we have enlarged the font size of axes and legends of Figure 5 and Figure 6.

[Figure]

**Figure 1 Humic index and Peak T/Peak C ratio served as indictors of humification degree and the biodegradable possibility of WSOC. (a) HIX in different particle sizes, large HIX value indicated high humification degree or high aromaticity of fluorescent organics. (b) Peak T/Peak C ratios of different particle sizes. The large value indicated more microbial metabolites in the fluorescent organics. (c) showed the size distributions of $\eta_{WH>320}$ for winter and summer samples, respectively.**

[Figure]

**Figure 2 PARAFAC results of EEM in winter and summer respectively. Three components were extracted of both seasons, the portions of each component for different particle sizes were shown as well.**

d. Figure 7: No x-axis label or units.

**Response:** We are sorry for the mistakes. The x-axis label is added, while the GRDs are of no unit, since they are calculated by normalized data.

[Figure]

**Figure 3 GRD of size segregated WSOC, AFI, and average UV. (a) and (b) GRD calculated**

by WSOC, AFI, and average UV of each sample, setting data of <0.26 μm as references, GRD (<0.26) =1; (c) and (d) GRD between WSOC and light absorption indices, setting WSOC as references.